# Stretched Exponential Convergence of (Stochastic) Gradient Descent for Separable Logistic Regression

**Sacchit Kale** *sacchitm@iisc.ac.in*
*Undergraduate Programme*
*Indian Institute of Science, Bangalore*

**Piyushi Manupriya** *piyushim@iisc.ac.in*
*Department of Computer Science and Automation*
*Indian Institute of Science, Bangalore*

**Pierre Marion** *pierre.marion@inria.fr*
*Inria, Ecole Normale Supérieure*
*PSL Research University, Paris, France*

**Francis Bach** *francis.bach@inria.fr*
*Inria, Ecole Normale Supérieure*
*PSL Research University, Paris, France*

**Anant Raj** *anantraj@iisc.ac.in*
*Department of Computer Science and Automation*
*Indian Institute of Science, Bangalore*

**Reviewed on OpenReview:** *https: // openreview. net/ forum? id=R5OaFwCmSO*

## Abstract

Gradient descent and stochastic gradient descent are central to modern machine learning, yet their behavior under large step sizes remains theoretically unclear. Recent work suggests that acceleration often arises near the edge of stability, where optimization trajectories become unstable and difficult to analyze. Existing results for separable logistic regression achieve faster convergence by explicitly leveraging such unstable regimes through constant or adaptive large step sizes. In this paper, we show that instability is not inherent to acceleration. We prove that gradient descent with a simple, non-adaptive increasing step-size schedule achieves stretched exponential convergence for separable logistic regression under a margin condition, while remaining entirely within a stable optimization regime. The resulting method is anytime and does not require prior knowledge of the optimization horizon or target accuracy. We also establish stretched exponential convergence of stochastic gradient descent using a lightweight adaptive step-size rule that avoids line search and specialized procedures, improving upon existing polynomial-rate guarantees. Together, our results demonstrate that carefully structured step-size growth alone suffices to obtain stretched exponential acceleration for both gradient descent and stochastic gradient descent.

## 1 Introduction

Gradient-based methods form the backbone of modern machine learning optimization. Algorithms such as gradient descent (GD), stochastic gradient descent (SGD), and their momentum-based variants are the most widely used tools for training large-scale models, owing to their conceptual simplicity, scalability, and empirical effectiveness. Consider an unconstrained optimization problem with objective $\mathcal{L} : \mathbb{R}^d \to \mathbb{R}$,

$$\min_{\mathbf{w} \in \mathbb{R}^d} \mathcal{L}(\mathbf{w}).$$

The gradient descent iteration for this problem is given by

$$\mathbf{w}_{t+1} = \mathbf{w}_t - \eta \nabla \mathcal{L}(\mathbf{w}_t),$$

where $\eta > 0$ denotes the step size (learning rate). When $\mathcal{L}$ is convex and $L$-smooth, the convergence properties of gradient descent and its variants are well understood and form a cornerstone of classical optimization theory (Nesterov et al., 2018; Garrigos & Gower, 2023).

Classical analyses dictate that stability and convergence of gradient descent require the step size to be sufficiently small, typically $\eta \leq 2/L$, where $L = \lambda_{\max}(\nabla^2 \mathcal{L})$ is the smoothness parameter. Similarly, stochastic gradient descent—where the true gradient is replaced by an unbiased stochastic approximation—is known to converge under diminishing step-size schedules (Robbins & Monro, 1951; Garrigos & Gower, 2023). These prescriptions have long guided both theory and practice.

However, the emergence of large-scale machine learning models has revealed a striking mismatch between theory and practice. Empirically, practitioners routinely employ learning rates that far exceed classical stability limits, yet both gradient descent and stochastic gradient descent continue to make progress, often driving the training loss down over long horizons. Recent works (Ahn et al., 2022; Cohen et al., 2021) have shown that, in many neural network training problems, full-batch gradient descent operates in a regime characterized by transient instability followed by sustained loss decrease. This phenomenon is now commonly referred to as the *edge of stability*. Despite its empirical prevalence, theoretical understanding of optimization dynamics in this regime remains limited (Arora et al., 2022).

Parallel to these empirical observations, another line of work has explored carefully designed learning-rate schedules that provably accelerate convergence beyond classical gradient descent rates (Altschuler & Parrilo, 2025). While these methods achieve strong theoretical guarantees, the resulting schedules are often intricate and significantly depart from the simple heuristics used in practice.

A breakthrough in understanding the benefits of large step sizes was provided by Wu et al. (2024), who demonstrated that gradient descent with a *very large constant step size* can achieve faster convergence for separable logistic regression. Given training data $\{(\mathbf{x}_i, y_i)\}_{i=1}^n$ with labels $y_i \in \{\pm 1\}$, consider the logistic loss

$$\mathcal{L}(\mathbf{w}) = \frac{1}{n} \sum_{i=1}^n \ln\big(1 + \exp(-y_i \mathbf{x}_i^\top \mathbf{w})\big).$$

Under the assumption that the data are linearly separable, i.e., there exists $\mathbf{w}^\star$ such that $y_i \mathbf{x}_i^\top \mathbf{w}^\star \geq \gamma$ for all $i$ and some margin $\gamma > 0$, Wu et al. (2024) proved that gradient descent with a sufficiently large constant step size converges at rate $\mathcal{O}(1/T^2)$ after an oscillating phase of $\mathcal{O}(T/2)$, improving upon the classical $\mathcal{O}(1/T)$ rate for smooth convex objectives. Subsequent works extended these results to obtain stretched exponential convergence rates using adaptive but large step-size schedules (Zhang et al., 2025b), and further analyzed large step-size gradient descent for regularized logistic regression (Wu et al., 2025).

A unifying theme across these works is that faster-than-classical convergence rates appear to be achievable only by passing through a *non-monotonic and unstable optimization phase*. Theoretical analyses therefore require a delicate decomposition of the optimization trajectory into multiple regimes, including an initial edge-of-stability phase, a transition period, and a final stable phase. This complexity has limited the scope and generality of existing results. For stochastic gradient descent, although large learning rates were shown to provide benefits by Wu et al. (2024), the improvements were substantially weaker than those observed for full-batch gradient descent.

In this work, we take a fundamentally different approach. We show that fast convergence of separable logistic regression can be achieved *without entering the edge-of-stability regime*. Specifically, we prove that gradient descent with a *simple, non-adaptive increasing step-size schedule* achieves stretched exponential convergence for separable logistic regression. This rate is strictly faster than the polynomial rates established by Wu et al. (2024) and is close to the exponential rates obtained by Zhang et al. (2025b), while being *anytime* and avoiding adaptive or complex learning-rate schedules.

We also analyze stochastic gradient descent in the separable logistic regression setting. Inspired by the techniques of Zhang et al. (2025b), we establish stretched exponential convergence of *vanilla SGD* for sepa-

rable logistic regression under increasing adaptive step sizes. To the best of our knowledge, this is the first result showing stretched exponential convergence for SGD in this setting without resorting to line search or specialized adaptive procedures. Recent work by Vaswani & Babanezhad (2025) establishes stretched exponential convergence of SGD using an Armijo line search. However, the proof in the published ICML version contains a technical issue arising from improper conditioning on future randomness. A corrected analysis addressing this issue appeared in a subsequent revision released independently and contemporaneously with this work. In contrast, our results show that standard SGD with increasing (and potentially large) step sizes attains comparable guarantees over a broader tolerance regime, without requiring any line search and without knowledge of the final tolerance level. Moreover, by conditioning on the hitting time, which is adapted to the filtration, we obtain a corrected analysis that avoids any dependence on future randomness and relies solely on events measurable with respect to the filtration. Taken together, our results show that instability is not a prerequisite for acceleration. Carefully structured yet simple step-size growth suffices to achieve stretched exponential convergence for both gradient descent and stochastic gradient descent in separable logistic regression.

**Contributions.** To this end, we make the following contributions in this paper:

1. **Anytime stretched exponential convergence of gradient descent (Section 3.2).** We establish an anytime stretched exponential convergence rate for gradient descent on separable logistic regression under a margin assumption, using a simple non-adaptive, increasing step-size schedule. The method does not require prior knowledge of the optimization horizon or target accuracy. In contrast to existing analyses achieving faster-than-$\mathcal{O}(1/T)$ rates via constant or adaptive large step sizes (Wu et al., 2024; Zhang et al., 2025b), our approach avoids unstable or edge-of-stability regimes altogether. The optimization trajectory remains globally stable throughout, showing that stretched exponential convergence can be achieved with large step sizes without incurring transient instability.

2. **Stretched exponential convergence of stochastic gradient descent with adaptive step-size (Section 3.3, Section 3.4)**

   We show that stochastic gradient descent achieves stretched exponential convergence for separable logistic regression under a margin condition using a lightweight adaptive step-size schedule. Adaptive SGD method **(Section 3.3)** depends on the observed stochastic loss and target tolerance level. We remove the need to know the tolerance level a priori via a Block Adaptive SGD scheme **(Section 3.4)**. This comes at the cost of an additional logarithmic factor in the overall complexity, while still preserving stretched exponential convergence. The proposed Adaptive SGD schemes do not rely on line search or prior knowledge of the target accuracy, and depends only on the observed stochastic loss at each iteration. This leads to strictly faster rates than the polynomial convergence guarantees established in Wu et al. (2024).

**Organization of the paper:** The remainder of the paper is organized as follows. Section (2) reviews related literature. Section (3) presents our main theoretical results, first for gradient descent and then for stochastic gradient descent in separable logistic regression. Section (4) reports empirical results, and Section (5) concludes the paper.

## 2 Related Works

Classical analyses of gradient descent (GD) for smooth convex loss functions typically employ a fixed step-size $\eta$ as inverse of the global smoothness constant. This choice guarantees monotonic descent of the loss but leads only to sub-linear convergence rate of $\mathcal{O}(1/T)$, with $T$ as the horizon. Obtaining linear convergence rate generally requires the loss $\mathcal{L}(\cdot)$ to be strongly convex. Line-search (backtracking) methods (Vaswani & Babanezhad, 2025) adaptively tune the step size to guarantee monotonic descent, but they typically introduce additional per-iteration computational overhead. Moreover, the current SGD analysis in Vaswani & Babanezhad (2025) contains a minor flaw in the published ICML version which has been now fixed in a revised version contemporaneously and independently. To adapt to the local geometry, Polyak scheme

(Polyak, 1969) chooses variable step-sizes, $\eta_t := \frac{\mathcal{L}(\mathbf{w}_t) - \mathcal{L}^\star}{\|\nabla \mathcal{L}(\mathbf{w}_t)\|^2}$, which requires the knowledge of $\mathcal{L}^\star$ and only results in $\mathcal{O}(1/T)$ convergence rate for smooth convex functions (He et al., 2025). Logistic loss, a smooth convex function widely used in classification, has motivated a growing body of work aimed at providing rigorous theoretical justification for its empirical success. More recent work has demonstrated that exploiting its local smoothness properties can substantially accelerate convergence. In particular, for logistic regression under linearly separable data, Axiotis & Sviridenko (2023) result in near exponential convergence (after a transient time) by employing $\eta_t \leq \min\left\{\frac{1}{2\mu\mathcal{L}(\mathbf{w}_t)}, \frac{1}{\bar{\gamma}\|\nabla\mathcal{L}(\mathbf{w}_t)\|}\right\}$, where $\mu$ and $\bar{\gamma}$ are the constants associated with the local geometric properties of second-order robustness and multiplicative smoothness defined by Axiotis & Sviridenko (2023). A related but distinct line of research studies adaptive first-order methods that automatically tune step-sizes without explicit curvature information and obtain exponential convergence rate (Malitsky & Mishchenko, 2020) but with a dependence on the local strong convexity parameter. As large step-sizes for logistic loss with linearly separable data often outperform in practice, recent studies like Wu et al. (2024) have shown improved convergence rates of the order $\mathcal{O}(1/(\eta T))$, where $\eta = \mathcal{O}(T)$, after an initial *unstable* loss-oscillation phase lasting upto $T/2$ iterations. Such an analysis was extended showing improved convergence rate with adaptive step-sizes by Zhang et al. (2025a;b) which proves exponential rates after a burn-in time of $1/\gamma^2$. Before reaching the burn-in time, Zhang et al. (2025a;b) only provide an upper bound on the loss at the average iterate and for $t < 1/\gamma^2$, this upper bound is of the form $\mathcal{L}(\bar{\mathbf{w}}_t) \leq \exp(c\eta/t)$, where $\bar{\mathbf{w}}_t$ is the average iterate at timestep $t$ and $c > 0$.

To the best of our knowledge, such stretched exponential convergence rates for GD with large and non-adaptive step-sizes are not known. Moreover, these proof techniques do not extend straightforwardly to stochastic gradient descent (SGD). Wu et al. (2024) only discuss the initial unstable bound in SGD for logistic regression and give high probability bounds on the average loss as $O(\eta/T)$. The literature related to step-sizes adapted to local smoothness for SGD is also relatively sparse.

## 3 Gradient Descent Schemes for Logistic Loss

It is by now well understood that gradient descent for logistic regression with linearly separable data exhibits non-standard convergence behavior. In particular, the loss converges to zero while the parameter norm diverges, and the optimization trajectory implicitly aligns with the maximum-margin direction (Soudry et al., 2018; Ji & Telgarsky, 2018; Nacson et al., 2019a; Ji & Telgarsky, 2021; Wu et al., 2023). Achieving fast (e.g., linear or near-linear) rates for the loss typically relies on carefully controlling this trajectory.

Recent works establishing fast rates for logistic regression with large or adaptive step-sizes follow a *two-phase analysis* (Wu et al., 2024; Zhang et al., 2025a). In the first phase, often referred to as the *edge-of-stability* or *unstable phase*, the algorithm operates with aggressive step-sizes that induce oscillations or non-monotonicity in the loss. This transient instability is essential in these analyses to rapidly increase the norm of the iterate and move the algorithm into a favorable region of the parameter space. Once this region is reached, the dynamics transition into a second, *stable phase*, where the loss decreases monotonically and fast convergence rates can be proved.

While successful, such two-phase analyses are inherently delicate. They rely on explicitly characterizing when instability occurs, when it ends, and how the algorithm transitions into the stable regime. Moreover, instability is typically an unavoidable feature of these methods: large step-sizes are introduced early, and loss oscillations are a necessary byproduct of accelerating progress.

In contrast, we show that fast convergence for logistic regression can be achieved using a simple, non-adaptive *increasing step-size scheme* that never enters an unstable regime. Despite step-sizes growing over time, the optimization trajectory remains globally controlled, and loss instability is avoided altogether.

The convergence behavior of stochastic gradient descent (SGD) for logistic loss with linearly separable data has been shown to be similar to that of gradient descent (Nacson et al., 2019b) for the classical choice of small step-size. However, the stochastic setting presents additional challenges that are not readily captured by existing analyses of gradient descent. For SGD, fast or exponential convergence guarantees with large step-size remain far less understood. A recent work by Umeda & Iiduka (2025) propose increasing step-size

scheme for SGD but do not specialize to logistic loss and do not obtain exponential convergence rate. The results in Wu et al. (2024) only provide high probability bounds of the order $\mathcal{O}(\eta/T)$ on the average loss in the initial unstable phase. In particular, the two-phase analyses that underpin fast rates in the deterministic setting — where an initial unstable phase drives rapid growth of the iterate norm followed by a stable phase with monotone loss decrease — rely crucially on deterministic control of the optimization trajectory and do not carry over to stochastic updates. Our work departs from the two stage analysis with large step-size and rather designs local curvature based step-size for SGD that obtains a near exponential convergence rate.

### 3.1 Preliminaries

**Notations.** We use boldface to refer to vectors/matrices. With dimensions depending on the context, $\mathbf{I}$ refers to the Identity matrix. $\mathbf{1}\{\cdot\}$ refers to the binary indicator random variable. All norms are $\ell_2$ norms unless specified otherwise.

**Properties of the Logistic Loss.** We consider the empirical logistic loss

$$\mathcal{L}(\mathbf{w}) := \frac{1}{n} \sum_{i=1}^{n} \ln\left(1 + \exp(-y_i \mathbf{x}_i^\top \mathbf{w})\right), \tag{1}$$

where $(\mathbf{x}_i, y_i) \in \mathbb{R}^d \times \{\pm 1\}$.

We begin by discussing some standard properties of logistic loss that were presented in Wu et al. (2024). As a differentiable convex function, logistic loss $\mathcal{L}(\cdot)$ satisfies $\mathcal{L}(\mathbf{w}_1) \geq \mathcal{L}(\mathbf{w}_2) + \nabla \mathcal{L}(\mathbf{w}_2)^\top (\mathbf{w}_1 - \mathbf{w}_2)$ for any $\mathbf{w}_1, \mathbf{w}_2 \in \mathbb{R}^d$.

Throughout, we impose the standard separability assumption.

**Assumption 3.1.** The training data satisfy:

1. (*Linear separability*) There exists a unit vector $\mathbf{w}^\star$ and $\gamma > 0$ such that $y_i \mathbf{x}_i^\top \mathbf{w}^\star \geq \gamma$ for all $i \in [1, n]$.

2. (*Normalized features*) $\|\mathbf{x}_i\| \leq 1$ for all $i \in [1, n]$.

A crucial property that the logistic loss satisfies under Assumption (3.1) is the loss controlling the maximum eigenvalue of the Hessian. Let $\mathbf{H}(\mathbf{w})$ denote the Hessian for the logistic loss computed at $\mathbf{w}$ and $\lambda_j$'s as the eigenvalues of the (symmetric) Hessian matrix $\mathbf{H}(\cdot)$, $\lambda_{\max}(\mathbf{H}(\mathbf{w})) \leq \sqrt{\sum_{j=1}^{d} \lambda_j^2} = \|\mathbf{H}\|_F$.

$$\lambda_{\max}(\mathbf{H}(\mathbf{w})) \leq \left\| \sum_{i=1}^{n} \frac{1}{n} \mathbf{x}_i \mathbf{x}_i^\top \frac{\exp\left(-y_i \mathbf{x}_i^\top \mathbf{w}\right)}{(1 + \exp\left(-y_i \mathbf{x}_i^\top \mathbf{w}\right))^2} \right\|_F \leq \frac{1}{n} \sum_{i=1}^{n} \|\mathbf{x}_i \mathbf{x}_i^\top\|_F \frac{\exp\left(-y_i \mathbf{x}_i^\top \mathbf{w}\right)}{(1 + \exp\left(-y_i \mathbf{x}_i^\top \mathbf{w}\right))^2}$$

$$\overset{\text{Assumption (3.1)}}{\leq} \frac{1}{n} \sum_{i=1}^{n} \frac{\exp\left(-y_i \mathbf{x}_i^\top \mathbf{w}\right)}{(1 + \exp\left(-y_i \mathbf{x}_i^\top \mathbf{w}\right))^2} \leq \min\left\{1/4, \mathcal{L}(\mathbf{w})\right\}. \tag{2}$$

From the above simplification, we also know that $\mathcal{L}(\mathbf{w})$ is a smooth function with a global smoothness constant $L = 1/4$.

This result also gives 1-smoothness of log of the logistic loss:

$$\nabla^2 \ln \mathcal{L}(\mathbf{w}) = \frac{1}{\mathcal{L}(\mathbf{w})^2} \left(\mathbf{H}(\mathbf{w}) \cdot \mathcal{L}(\mathbf{w}) - (\nabla \mathcal{L}(\mathbf{w}))^{\otimes 2}\right) \preccurlyeq \frac{1}{\mathcal{L}(\mathbf{w})} \mathbf{H}(\mathbf{w}) \overset{\text{From } equation \ 2}{\preccurlyeq} \frac{1}{\mathcal{L}(\mathbf{w})} \mathcal{L}(\mathbf{w}) \cdot \mathbf{I} = \mathbf{I}. \tag{3}$$

Under the linear separability given by Assumption (3.1), defining a comparator $\mathbf{u} := \frac{\beta}{\gamma} \mathbf{w}^\star$ for $\beta > 0$, we have the following.

$$\mathcal{L}(\mathbf{u}) \leq \frac{1}{n} \sum_{i=1}^{n} \exp\left(-y_i \mathbf{x}_i^\top \mathbf{w}^\star (\beta/\gamma)\right) \leq \frac{1}{n} \sum_{i=1}^{n} \exp(-\beta). \tag{4}$$

Finally, the self-bounded gradient property also holds with normalized features Assumption (3.1):

$$\|\nabla\mathcal{L}(\mathbf{w})\| = \left\|\frac{1}{n}\sum_{i=1}^{n}\frac{1}{1+\exp\left(y_i\mathbf{x}_i^\top\mathbf{w}\right)}(-y_i\mathbf{x}_i)\right\| \leq \frac{1}{n}\sum_{i=1}^{n}\frac{\|y_i\mathbf{x}_i\|}{1+\exp\left(y_i\mathbf{x}_i^\top\mathbf{w}\right)} \leq \min\left\{1,\mathcal{L}(\mathbf{w})\right\}. \tag{5}$$

Now, we discuss our step-size scheme and the corresponding analysis for gradient descent and stochastic gradient descent.

## 3.2 Analysis with Gradient Descent

As motivated in previous sections, classical analysis of gradient descent (GD) for smooth convex objectives using a fixed step-size chosen proportional to the inverse of a global smoothness constant often leads to slow convergence. Using larger constant step-sizes requires specifying the time-horizon ($T$) and can also induce loss oscillations up to a reasonably long duration of up to ($T/2$) (Wu et al., 2024; Zhang et al., 2025a).

These observations raise a natural question: *can one design a non-adaptive step size schedule that avoids oscillatory behavior without the computational overhead of line-search while achieving substantially faster convergence?*

In this section, we answer this question affirmatively. We propose a new deterministic step-size scheme for GD that is fully specified in advance, prevents loss oscillations, and yields convergence that is any time nearly exponential. We first observe that a desirable property of the step-size $\eta_t$ to showcase faster convergence while avoiding loss oscillations is to incorporate the local geometry. Logistic loss with separable data exhibits a self-bounding curvature property, with the maximum eigenvalue of Hessian controlled by the loss equation 2. This property was leveraged by Axiotis & Sviridenko (2023); Zhang et al. (2025a) who proposed adaptive step-size schemes. Our work removes the need to for tuning the step-sizes adaptively that comes with more intricate analysis. Instead, the proposed step-size scheme implicitly incorporates the local curvature and consequently ensures monotonically non-increasing loss values.
Consider the following gradient descent (GD) update rule for logistic loss:

$$\mathbf{w}_0 = \mathbf{0}$$
$$\mathbf{w}_{t+1} = \mathbf{w}_t - \eta_t\nabla\mathcal{L}(\mathbf{w}_t),\ t \geq 0. \tag{6}$$

With a general initialization $\mathbf{w}_0$, our proposed step-size $\eta_t$ is presented as follows

$$\eta_t := \begin{cases} \frac{1}{\ln(2)+\|\mathbf{w}_0\|} & ,\ t = 0 \\ \frac{S_{t-1}}{2\max\{2F(\mathbf{w}_0),\ \ln^2(S_{t-1})\}} & ,\ t > 0, \end{cases} \tag{7}$$

where $S_t := \gamma^2\sum_{k=0}^{t}\eta_k$, $\gamma$ is the margin of data separation and $F(\mathbf{w}_0) := \frac{1}{n}\sum_{i=1}^{n}\exp(-y_i\mathbf{x}_i^\top\mathbf{w}_0)$.
We highlight that the step-size scheme $\eta_t$ defined in equation 7 does not depend on local curvature or per-coordinate statistics and depends only on the initialization and $\gamma$. Its evolution is entirely deterministic and depends on a quantity $S_{t-1}$ that grows cumulatively.

In particular, during an initial phase up to a finite timestep $S_t$ follows a geometric growth rate. Following this phase, $\eta_t$ grows as $\frac{S_{t-1}}{2\ln^2(S_{t-1})}$. We show that unlike the case by Wu et al. (2024); Zhang et al. (2025a), our GD dynamics never enters an unstable phase of loss oscillations despite the step-sizes being large. This property follows from our choice of the step-size scheme that ensures $\mathcal{L}(\mathbf{w}_t) \leq \frac{1}{\eta_t}$ described as follows. Leveraging 1-smoothness of log of logistic loss equation 3, we first present a proof for the condition ensuring monotonic descent of the loss, originally discussed in the stable phase analysis of gradient descent with constant step-sizes Wu et al. (2024, Lemma 11), adapted to our setting of a variable step-size schedule. From 1-smoothness of log of logistic loss equation 3 and the GD update rule equation 6, we have that the condition $\mathcal{L}(\mathbf{w}_t) \leq 1/\eta_t\ \forall t \geq 0$, makes the loss values monotonically non-increasing:

$$\ln\left(\frac{\mathcal{L}(\mathbf{w}_{t+1}))}{\mathcal{L}(\mathbf{w}_t)}\right) \leq -\eta_t\|\nabla\mathcal{L}(\mathbf{w}_t)\|^2\left(\frac{1}{\mathcal{L}(\mathbf{w}_t)} - \frac{\eta_t}{2}\right) \leq 0, \tag{8}$$

where the last inequality uses $\mathcal{L}(\mathbf{w}_t) \leq 1/\eta_t$.

Furthermore, this property enables us to adapt the desirable upper bounds on the loss derived by Wu et al. (2024) for GD dynamics with constant step-size during the *stable phase*. We present this result below and defer our adapted proof to Appendix (B).

**Lemma 3.2.** *(Adapted from Wu et al. (2024, Lemma 11)) Suppose* $\mathcal{L}(\mathbf{w}_k) \leq \frac{1}{\eta_k}$ *holds* $\forall k \in [s, t-1]$ *for logistic loss, under Assumption (3.1), we have*

$$\mathcal{L}(\mathbf{w}_t) \leq \frac{2F(\mathbf{w}_s) + \ln^2(\gamma^2 \sum_{k=s}^{t-1} \eta_k)}{\gamma^2 \sum_{k=s}^{t-1} \eta_k},$$

*where* $F(\mathbf{w}_s) := \frac{1}{n} \sum_{i=1}^{n} \exp(-y_i \mathbf{x}_i^\top \mathbf{w}_s)$.

A careful application of this lemma allows us to derive stretched exponential convergence guarantees for gradient descent on the logistic loss as discussed below.

**Theorem 3.3.** *Consider the GD update rule equation 6 with the proposed step-size for logistic regression and consider initialization* $\mathbf{w}_0 = \mathbf{0}$. *Under Assumption (3.1), we prove the following*

1. $\forall t \geq 0$, $\mathcal{L}(\mathbf{w}_t) \leq \frac{1}{\eta_t}$ *ensuring monotonically decreasing loss iterates.*

2. *There exist constants* $c, C > 0$ *depending only on* $\gamma$ *and the initialization such that* $\forall t \geq 1$,

$$\mathcal{L}(\mathbf{w}_t) \leq \frac{Ct^{2/3}}{\exp(ct^{1/3})} = \exp(-\Omega(t^{1/3})). \tag{9}$$

We provide the proof in Appendix (A.1) with a proof sketch as follows.

*Proof Sketch.* The argument consists of two conceptual steps: a uniform inductive control of the loss along the trajectory, followed by a quantitative analysis of the resulting growth dynamics.

We first show that $\mathcal{L}(\mathbf{w}_t) \leq \frac{1}{\eta_t}$ for all $t \geq 0$, using strong induction. The base case holds by construction of $\eta_0$ together with the elementary bound $\mathcal{L}(\mathbf{w}) \leq \ln(2) + \|\mathbf{w}\|$. For the inductive step, assume $\mathcal{L}(\mathbf{w}_k) \leq 1/\eta_k$ for all $k \leq t-1$. Invoking Lemma (3.2) with $s = 0$, we upper bound $\mathcal{L}(\mathbf{w}_t)$ by an explicit function of $S_{t-1} := \gamma^2 \sum_{k=0}^{t-1} \eta_k$, involving only $S_{t-1}$ and $\ln^2(S_{t-1})$. The step-size sequence $\{\eta_t\}$ is chosen precisely so that this upper bound equals $1/\eta_t$, thereby closing the induction. Notably, this argument does not rely on any line-search or adaptive acceptance criterion, but instead exploits the self-consistency between the loss bound and the step-size schedule.

We next analyze the evolution of $S_t$, which directly determines the loss decay via the bound established above. By definition,

$$S_t = S_{t-1} + \gamma^2 \eta_t.$$

Under the chosen step-size rule, this yields the nonlinear recurrence

$$S_t = S_{t-1} \left( 1 + \frac{\gamma^2}{2 \max\{2F(\mathbf{w}_0), \ln^2(S_{t-1})\}} \right).$$

Till a finite amount of time the recurrence has geometric growth. Afterwards the dynamics are governed by

$$S_t = S_{t-1} \left( 1 + \frac{\gamma^2}{2 \ln^2(S_{t-1})} \right).$$

Lemmas (A.1) and (A.2) together imply that in this regime

$$\ln(S_t) = \Theta(t^{1/3}), \quad \text{equivalently,} \quad S_t = \exp(\Theta(t^{1/3})).$$

Combining this growth estimate with the inductive loss bound yields the stated convergence rate for $\mathcal{L}(\mathbf{w}_t)$. □

### 3.3 Analysis with Stochastic Gradient Descent

Our next contribution demonstrates the role of local curvature based step-sizes to enable stretched exponential convergence with stochastic gradient descent (SGD). We analyze the convergence of the SGD update rule

$$\mathbf{w}_0 = \mathbf{0}$$
$$\mathbf{w}_{t+1} = \mathbf{w}_t - \eta_t \nabla \mathcal{L}_{i_t}(\mathbf{w}_t), \ t \geq 0, \tag{10}$$

where $\mathcal{L}_{i_t}(\mathbf{w}_t) \coloneqq \ln\left(1 + \exp\left(-y_{i_t} \mathbf{x}_{i_t}^\top \mathbf{w}_t\right)\right)$ and the index $i_t \in [1, n]$ is chosen uniformly at random.

As in the GD case, our choice of step-size is guided by the structural property that the curvature of the logistic loss is controlled by its value, which suggests that large step-sizes can be taken safely in regions of low loss. We propose the step-size scheme defined as

$$\eta_t \coloneqq \min\left\{\frac{1}{\varepsilon}, \ \frac{1}{\mathcal{L}_{i_t}(\mathbf{w}_t)}\right\}, \tag{11}$$

where $\varepsilon$ is the final tolerance level/optimality gap. Henceforth, we will refer to this update rule as Adaptive SGD.

This proposed step-size can be seen as stochastic analogue of the step-size proposed by Axiotis & Sviridenko (2023) for GD. However, we note that the analysis differs significantly from that of Axiotis & Sviridenko (2023). In the stochastic setting, instability may arise from occasional noisy gradients. Hence, unlike GD, the SGD dynamics is not expected to exhibit monotonic descent even when analyzing the expected loss. This calls for a different proof technique. We begin by defining

$$\tau \coloneqq \inf\{t \geq 0 : \ \mathcal{L}(\mathbf{w}_t) \leq \varepsilon\}. \tag{12}$$

Let the canonical filtration be
$$\mathcal{F}_t \coloneqq \sigma(i_0, \ldots, i_{t-1}), \qquad t \geq 0, \tag{13}$$

with $\mathcal{F}_0 = \{\emptyset, \Omega\}$. Since the update is deterministic given $(i_0, \ldots, i_{t-1})$ and the data, $\mathbf{w}_t$ is $\mathcal{F}_t$-measurable for every $t$, hence $\mathcal{L}(\mathbf{w}_t)$ is $\mathcal{F}_t$-measurable. Therefore,

$$\{\tau \leq t\} = \{\mathcal{L}(\mathbf{w}_t) \leq \varepsilon\} \in \mathcal{F}_t,$$

so $\tau$ is a stopping time. We note that even when conditioning on the event that the algorithm has not yet reached the target accuracy, the sampling distribution of the stochastic index remains uniform. This holds because the pre-hitting event, $\{\tau < t\}$, is measurable with respect to the past filtration $(\mathcal{F}_t)$ and therefore does not bias the independent sampling process. We provide a formal proof for this in Lemma (A.5). Further, following the non-negativity of logistic loss, conditioned on the event that the stopping condition has not been reached ($\{t < \tau\}$), there must exist at least one datapoint whose loss remains larger than a threshold (Lemma (A.6)). This observation provides the key lower bound on progress used in the analysis.

We now state our main convergence result for Adaptive SGD applied to logistic regression under separability. The theorem establishes a finite expected hitting-time bound for reaching an $\varepsilon$-suboptimal iterate.

**Theorem 3.4.** *Under Assumption (3.1), Adaptive SGD equation 10 on logistic regression with the proposed step-size satisfies,*

$$\mathbb{E}[\tau] \ \leq \ \frac{2n}{\gamma^2} \ln^2\left(\frac{4n}{\varepsilon}\right).$$

*Consequently, for any $\delta \in (0, 1)$,*

$$\mathbb{P}\left(\tau < \frac{1}{\delta} \cdot \frac{2n}{\gamma^2} \ln^2\left(\frac{4n}{\varepsilon}\right)\right) \geq 1 - \delta.$$

We provide the proof of this result in Appendix (A.2). We also give a proof sketch below.

*Proof Sketch.* We consider the potential $D_t = \|\mathbf{w}_t - \mathbf{u}\|^2 \mathbf{1}\{t < \tau\}$ for a comparator $\mathbf{u} = \frac{\mathbf{w}^\star}{\gamma} \ln\left(\frac{4n}{\epsilon}\right)$. Before the hitting time $\tau$, Lemma (A.8) shows that the process has a uniform negative drift,

$$\mathbb{E}[D_{t+1} \mid \mathcal{F}_t] \leq D_t - \frac{1}{2n}.$$

A telescoping argument (Lemma (A.7)) then yields $\mathbb{E}[\tau] \leq 2n \|\mathbf{w}_0 - \mathbf{u}\|^2$. From this we obtain,

$$\mathbb{E}[\tau] \leq \frac{2n}{\gamma^2} \ln^2\left(\frac{4n}{\varepsilon}\right).$$

Finally, applying Markov's inequality converts the bound on the expectation into the stated high-probability guarantee. We note that this proof also applies to a broader class of non-negative smooth convex binary classification losses with an exponential tail whose gradients have self-bounding property ($\|\nabla\mathcal{L}(\cdot)\| \leq \mathcal{L}(\cdot)$). $\qquad\square$

## 3.4 Analysis with Block Adaptive SGD

Adaptive SGD equation 10, with step-size $\eta_t = \min\left\{\frac{1}{\varepsilon}, \frac{1}{\mathcal{L}_{i_t}(\mathbf{w}_t)}\right\}$, requires prior knowledge of the target tolerance level $\varepsilon$. This dependency can be removed using a doubling-trick strategy. To this end, we propose a *Block Adaptive SGD* algorithm that progressively refines the effective tolerance without requiring it as an input parameter. The algorithm is described as follows.

Fix $\varepsilon_0 \in (0, 1)$ and define $\varepsilon_k = \varepsilon_0/2^k$, $s_0 = 0$, $s_{k+1} = s_k + N_k$ for a block length $N_k (> 0)$. During block $k$ (iterations $t \in \{s_k, \ldots, s_{k+1} - 1\}$), update

$$\mathbf{w}_{t+1} = \mathbf{w}_t - \eta_t \nabla\mathcal{L}_{i_t}(\mathbf{w}_t), \qquad \eta_t = \min\left\{\frac{1}{\varepsilon_k}, \frac{1}{\mathcal{L}_{i_t}(\mathbf{w}_t)}\right\}, \tag{14}$$

where $i_t$ is chosen uniformly at random. For a target $\varepsilon \in (0, \varepsilon_0]$, define the activated block

$$k_\varepsilon = \min\{k : \varepsilon_k \leq \varepsilon\}.$$

The Block Adaptive SGD algorithm proceeds in stages (blocks) with progressively decreasing target accuracy levels $\varepsilon_k$. Within each block, stochastic gradient descent is run with an adaptive step-size that scales inversely with the current loss, but is capped by $1/\varepsilon_k$. As the algorithm moves to later blocks, the step-size cap increases, enabling more aggressive updates and driving the loss toward smaller target levels. The activated block $k_\varepsilon$ corresponds to the first stage whose target accuracy is lower than the desired tolerance $\varepsilon$.

The following result establishes an anytime convergence guarantee for Block Adaptive SGD obtained via the doubling-trick schedule.

**Theorem 3.5.** *Fix $\delta \in (0, 1)$, let $\varepsilon_k := \frac{\varepsilon_0}{2^k}$, $N_k := \left\lceil \frac{4n}{\delta\gamma^2} \ln^2\left(\frac{8n}{\delta\varepsilon_k}\right) \right\rceil$. Let $\varepsilon \in (0, \varepsilon_0]$ be a target accuracy and define $k_\varepsilon := \min\{k : \varepsilon_k \leq \varepsilon\}$, $\bar{\varepsilon} := \varepsilon_{k_\varepsilon}$. Define the post-activation hitting time $\tau := \inf\{t \geq s_{k_\varepsilon} : \mathcal{L}(\mathbf{w}_t) \leq \bar{\varepsilon}\}$.*

*Under Assumption (3.1), Block Adaptive SGD equation 14 applied to logistic regression satisfies:*

(i) *(**Probability bound**) One has $\mathbb{P}\left(\min_{0 \leq t \leq s_{k_\varepsilon+1}} \mathcal{L}(\mathbf{w}_t) \leq \varepsilon\right) \geq 1 - \delta$.*

(ii) *(**Expected hitting time inside the activated block**) Let $N := N_{k_\varepsilon}$. Then almost surely,*

$$\mathbb{E}[(\tau - s_{k_\varepsilon})_+ \wedge N] = \mathcal{O}\left(\frac{n}{\gamma^2} \ln^2\left(\frac{n}{\delta\varepsilon}\right)\right).$$

(iii) *(**Total iteration complexity**) The total number of iterations up to the end of the activated block satisfies*

$$s_{k_\varepsilon+1} = \mathcal{O}\left(\frac{n}{\delta\gamma^2} \ln^3\left(\frac{n}{\delta\varepsilon}\right)\right).$$

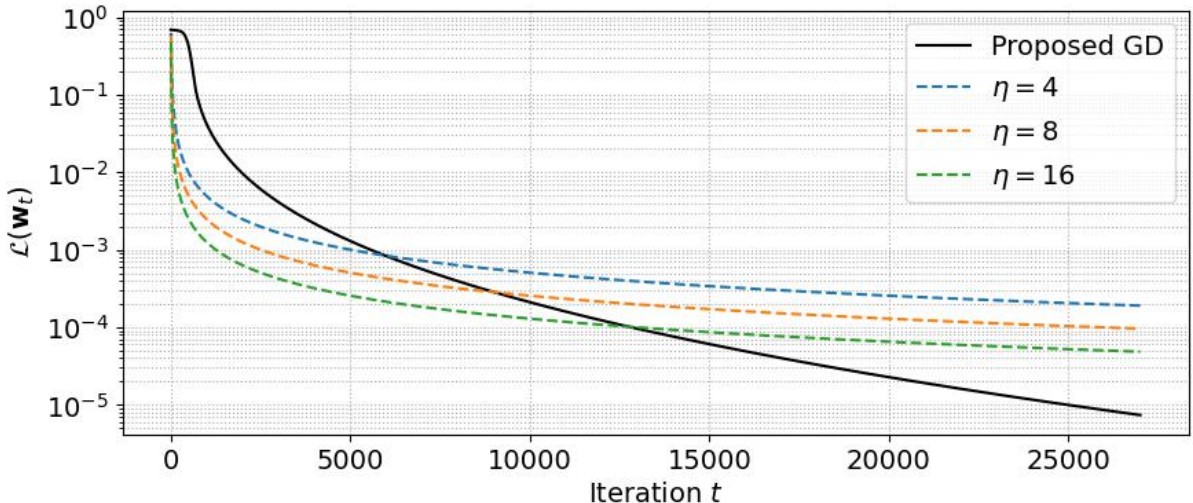

**Figure 1:** Comparison of our GD equation 6 and constant step-size gradient descent for logistic regression on a synthetic linearly separable dataset. The plot shows the evolution of the empirical logistic loss $\mathcal{L}(\mathbf{w}_t)$ (log scale) as a function of iterations $t$.

A complete proof is provided in Appendix (A.3), and a proof sketch is outlined below.

*Proof Sketch.* Fix the activated block index $k = k_\varepsilon$. We study the post-block-start hitting time

$$\tau = \inf\{t \geq s_k : \ \mathcal{L}(\mathbf{w}_t) \leq \varepsilon_k\}.$$

The proof is based on a drift argument for the squared distance to a scaled $\mathbf{w}^\star$ (3.1) comparator. Expanding the SGD update and using convexity of the logistic loss together with the adaptive step-size yields a pathwise inequality of the form

$$\|\mathbf{w}_{t+1} - \mathbf{u}\|^2 \leq \|\mathbf{w}_t - \mathbf{u}\|^2 - \eta_t \mathcal{L}_{i_t}(\mathbf{w}_t) + 2\eta_t \mathcal{L}_{i_t}(\mathbf{u}).$$

Choosing $\mathbf{u} = \frac{1}{\gamma}\ln\left(\frac{8n}{\delta\varepsilon_k}\right)\mathbf{w}^\star$ ensures $\mathcal{L}(\mathbf{u}) \leq \delta\varepsilon_k/(8n)$. Taking conditional expectations and using uniform sampling shows that, before the hitting time, the stopped process $\bar{\mathbf{w}}_t = \mathbf{w}_{t \wedge \tau}$ satisfies a uniform negative drift,

$$\mathbb{E}[\|\bar{\mathbf{w}}_{t+1} - \mathbf{u}\|^2 \mid \mathcal{F}_t] \leq \|\bar{\mathbf{w}}_t - \mathbf{u}\|^2 - \frac{1}{2n}.$$

Summing this inequality over the block gives a bound on the expected number of iterations before reaching the block target, $\mathbb{E}[(\tau - s_k)_+ \wedge N_k] \leq 2n \, \mathbb{E}\|\mathbf{w}_{s_k} - \mathbf{u}\|^2$. A separate growth control argument bounds $\mathbb{E}\|\mathbf{w}_{s_k} - \mathbf{u}\|^2$ using only the nonexpansive part of the update across previous blocks. Combining these estimates yields $\mathbb{P}(\tau > s_{k+1}) \leq \delta$ for the prescribed block length $N_k$. Since $\varepsilon_k \leq \varepsilon$, this implies the desired high-probability anytime guarantee, and the expectation bound follows from the same drift inequality. □

We summarize our analysis for GD and SGD by highlighting the role of large step-size in achieving faster convergence for logistic loss with linearly separable data. Our step-size for GD equation 7 gives a schedule based only on global problem parameters, without needing line search or adaptivity, giving stable descent of loss and resulting in near exponential convergence. Our step-size choice for SGD equation 11 based only on the stochastic loss and block tolerance parameter gives stretched exponential convergence. Finally, the Block Adaptive SGD equation 14 removes the requirement of prior knowledge of loss tolerance level.

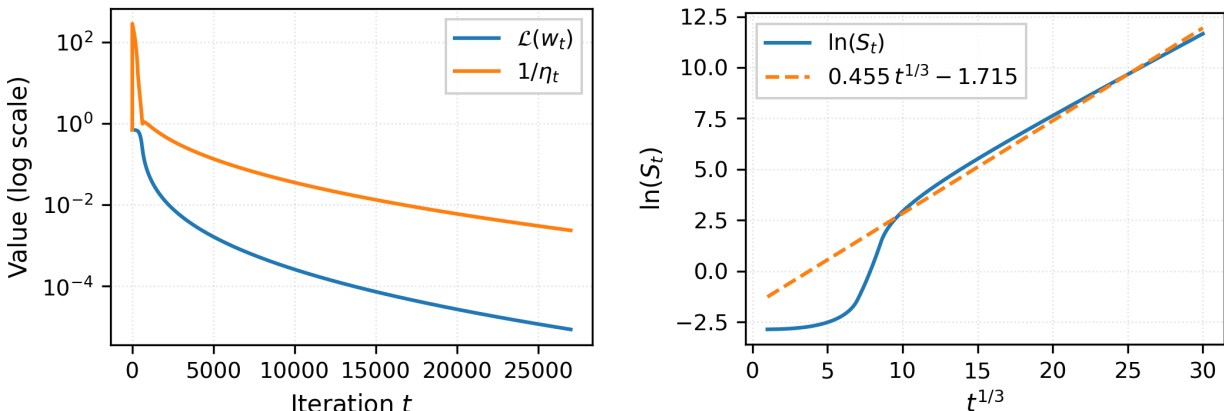

**Figure 2:** Dynamics of our GD equation 6 for logistic regression on a synthetic linearly separable dataset. **Left:** Evolution of the empirical loss $\mathcal{L}(\mathbf{w}_t)$ and inverse step size $1/\eta_t$ in log scale. **Right:** Plot of $\ln(S_t)$ versus $t^{1/3}$, validating order of growth of $\ln(S_t)$.

## 4 Experimental Results

In this section, we discuss empirical results on synthetic and real world datasets to verify our theoretical results from Section (3.2) and Section (3.3). Our plots are best viewed in color.

**Experiments for GD.** We study the GD dynamics for logistic regression on a synthetically generated 80-dimensional linearly separable dataset following Assumption (3.1) with $n = 1500$. GD is run with the proposed step-size scheme equation 7. Figure (2)-left shows the evolution of the empirical logistic loss $\mathcal{L}(\mathbf{w}_t)$ (in log scale) together with the inverse step size $1/\eta_t$ as a function of the iteration index $t$. Figure (2)-right plots $\ln(S_t)$ against $t^{1/3}$ and exhibits a linear trend. These empirical findings illustrate our theoretical results in Theorem (3.3).

In Figure (1), we show the improvement with the proposed GD update scheme compared to constant step-size GD, including the stepsize of $1/L$.

**Experiments for SGD.** We study the SGD dynamics for logistic regression on 10-dimensional synthetic data and a subset of the 1024-dimensional real world MNIST dataset (LeCun & Cortes, 2010) following Assumption (3.1). SGD is run with the proposed step-size scheme equation 11 for different values of $\varepsilon$. We plot the average empirical logistic loss (in log scale), obtained with 10 seeds that determine the choice of the randomly chosen data index in each SGD iteration, against square-root of the timescale. The synthetic data experiment has number of samples $n = 5000$ and shows stretched exponential convergence. The results are shown in Figure (4) which illustrates our theoretical claim of stretched exponential convergence in Theorem (3.4).

For the experiment with the MNIST dataset, we consider the task of classifying binary digits in the randomly sampled linearly separable subset of $n = 50$ samples. Figure (3) shows the results obtained for the classification experiments with different combination of digit pairs. We also plot the average loss with the quantile bands in Figure (6). Following an initial transient time, the losses show a sharp decrease exhibiting nearly linear trend in log-scale against $\sqrt{t}$. We further observe that smaller values of $\varepsilon$ delay the onset of this rapid descent but ultimately achieve significantly lower final loss values, whereas larger $\varepsilon$ leads to earlier but less pronounced convergence. This behavior is consistent with our theoretical predictions relating $\varepsilon$ to the achievable optimality gap and convergence rate. In Figure (5), we also show the empirically computed hitting times $\tau_\varepsilon$ as a function of $\varepsilon$ showing logarithmic dependence.

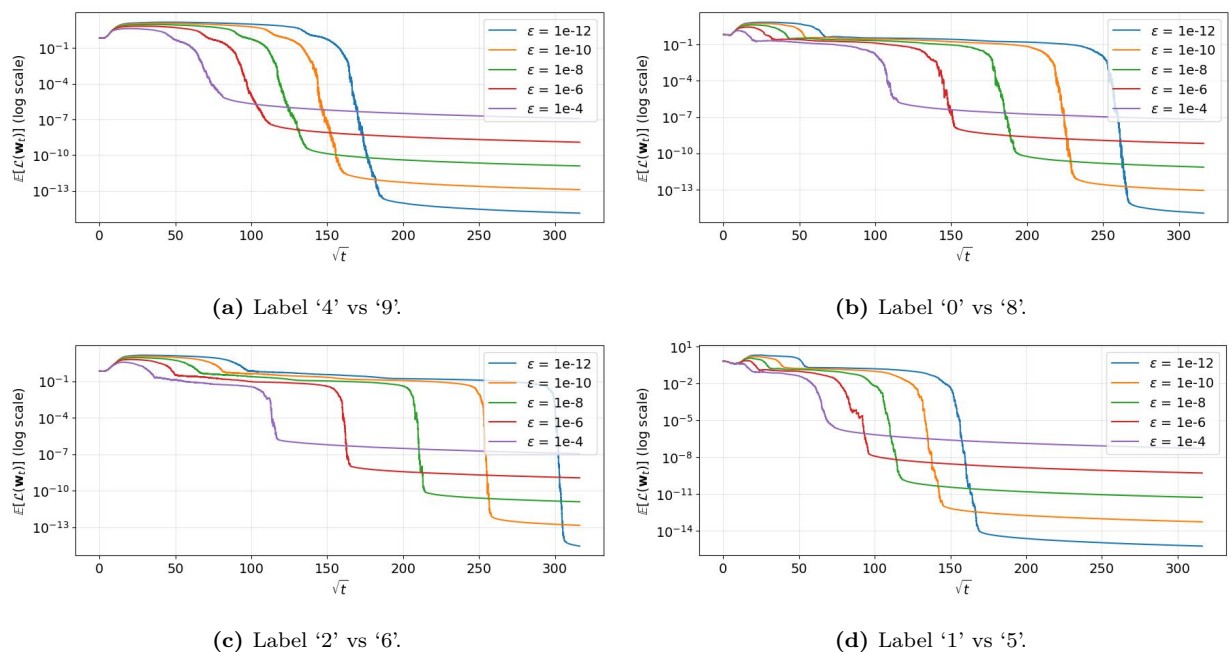

**Figure 3:** Average loss values with SGD equation 10 for logistic regression over an MNIST subset involving different pair of labels.

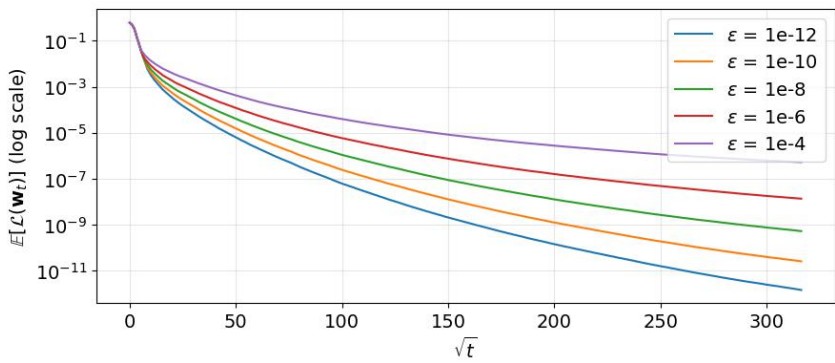

**Figure 4:** Average loss values with SGD equation 10 for logistic regression over a synthetically generated 10-dimensional data.

## 5    Conclusion and Future Work

Our first contribution demonstrates that the proposed choice of $\eta_t$ is pivotal in guaranteeing monotonic, non-increasing loss behavior with GD dynamics for logistic loss over separable data. Building on this, we establish an anytime stretched exponential convergence rate for gradient descent with logistic loss. The simplicity of our analysis offers a versatile framework that may inspire broader applications and may yield improved convergence guarantees for general loss functions.

We further analyze the SGD dynamics with a step-size that incorporates the local smoothness of the logistic loss and results in high probability bounds on the hitting time $\tau$. Our analysis for SGD is applicable to general convex non-negative binary classification losses with exponential tail that satisfy the self-bounded gradient property. We leave it as a future work to extend the analysis to a broader class of loss functions.

Our proposed Block Adaptive SGD provides a principled approach to remove the need to know the problem-specific tolerance level a priori. In practice, we found that the prescribed block sizes can be larger than what is needed in practice, which obscures its empirical advantage. We defer investigating empirical gains from Block Adaptive SGD as an interesting direction for future work.

**Acknowledgements**

Piyushi Manupriya was initially supported by a grant from Ittiam Systems Private Limited through the Ittiam Equitable AI Lab and then by ANRF-NPDF (PDF/2025/005277) grant. Anant Raj is supported by a grant from Ittiam Systems Private Limited through the Ittiam Equitable AI Lab, ANRF's Prime Minister Early Career Grant and Pratiksha Trust's Young Investigator Award. This work has received support from the French government, managed by the National Research Agency, under the France 2030 program with the reference "PR[AI]RIE-PSAI" (ANR-23-IACL-0008).

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

## A  Main Proofs

### A.1  Analysis with Gradient Descent

We recall the notations from Section (3.2). The GD update rule is given as

$$\mathbf{w}_0 = \mathbf{0}$$
$$\mathbf{w}_{t+1} = \mathbf{w}_t - \eta_t \nabla \mathcal{L}(\mathbf{w}_t), \ t \geq 0. \tag{15}$$

With a general initialization $\mathbf{w}_0$, our proposed step-size $\eta_t$ is presented as follows

$$\eta_t \coloneqq \begin{cases} \frac{1}{\ln(2) + \|\mathbf{w}_0\|} & , \ t = 0 \\ \frac{S_{t-1}}{2 \max\{2F(\mathbf{w}_0), \ \ln^2(S_{t-1})\}} & , \ t > 0, \end{cases} \tag{16}$$

where $S_t \coloneqq \left( \gamma^2 \sum_{k=0}^{t} \eta_k \right)$, $\gamma$ is the margin of data separation and $F(\mathbf{w}_0) \coloneqq \frac{1}{n} \sum_{i=1}^{n} \exp(-y_i \mathbf{x}_i^\top \mathbf{w}_0)$.

Lemma (A.1) establishes a lower bound for the recursion equation 17. It shows that $\ln^3(S_t)$ increases at least linearly with time, implying a stretched-exponential growth of $S_t$. The constant $C_1(s)$ captures the worst-case initial scaling and yields an explicit bound that will be used in the complexity analysis.

**Lemma A.1.** *(Lower bound on $S_t$.) Let $(S_t)_{t \geq s}$ satisfy*

$$S_t = S_{t-1} \left( 1 + \frac{\gamma^2}{2 \ln^2(S_{t-1})} \right), \quad t \geq s + 1, \tag{17}$$

*with $S_s > 1$. Define*

$$C_1(s) := 1 + \frac{\gamma^2}{2\ln^2(S_s)}.$$

*Then $\forall t \geq s+1$,*

$$\ln^3(S_t) \geq \ln^3(S_s) + \frac{3\gamma^2}{2C_1(s)}(t-s). \tag{18}$$

*Consequently, for all $t \geq s$,*

$$S_t \geq \exp\left(\left(\ln^3(S_s) + \frac{3\gamma^2}{2C_1(s)}(t-s)\right)^{1/3}\right). \tag{19}$$

*Proof.* From equation 17, for $k \geq s+1$

$$\ln(S_k) - \ln(S_{k-1}) = \ln\left(1 + \frac{\gamma^2}{2\ln^2(S_{k-1})}\right). \tag{20}$$

Using $\ln(1+x) \geq \frac{x}{1+x}$, $\forall x \geq 0$, we get

$$\ln(S_k) - \ln(S_{k-1}) \geq \frac{\gamma^2}{2\ln^2(S_{k-1}) + \gamma^2}. \tag{21}$$

Since $\ln(S_{k-1}) \geq \ln(S_s) > 0$ and

$$C_1(s) := 1 + \frac{\gamma^2}{2\ln^2(S_s)} \implies \gamma^2 = 2\big(C_1(s)-1\big)\ln^2(S_s) \leq 2\big(C_1(s)-1\big)\ln^2(S_{k-1}),$$

we have

$$2\ln^2(S_{k-1}) + \gamma^2 \leq 2C_1(s)\ln^2(S_{k-1}).$$

Hence, equation 21 gives

$$\ln(S_k) - \ln(S_{k-1}) \geq \frac{\gamma^2}{2C_1(s)\ln^2(S_{k-1})}$$

$$\implies \ln^2(S_{k-1})\big(\ln(S_k) - \ln(S_{k-1})\big) \geq \frac{\gamma^2}{2C_1(s)}. \tag{22}$$

Also, from the mean value theorem,

$$\ln^3(S_k) - \ln^3(S_{k-1}) = 3z^2\big(\ln(S_k) - \ln(S_{k-1})\big) \quad \text{for some } z \in [\ln(S_{k-1}), \ln(S_k)]$$
$$\geq 3\ln^2(S_{k-1})\big(\ln(S_k) - \ln(S_{k-1})\big). \tag{23}$$

Combining with equation 22, we obtain

$$\ln^3(S_k) - \ln^3(S_{k-1}) \geq \frac{3\gamma^2}{2C_1(s)}. \tag{24}$$

Summing from $s+1$ to $t$,

$$\ln^3(S_t) \geq \ln^3(S_s) + \frac{3\gamma^2}{2C_1(s)}(t-s), \tag{25}$$

where the inequality follows since $(\ln(S_t))$ is increasing.

The lower bound on $S_t$ follows by exponentiating. $\qquad\square$

Lemma (A.2) provides a upper growth bound for the recursion equation 26 . It shows that $\ln^3(S_t)$ grows at most linearly in time, yielding a stretched-exponential upper bound on $S_t$. Together with Lemma (A.1), this tightly characterizes the growth rate of the sequence.

**Lemma A.2.** *(Upper bound on $S_t$.) Let $(S_t)_{t \geq s}$ satisfy*

$$S_t = S_{t-1}\left(1 + \frac{\gamma^2}{2\ln^2(S_{t-1})}\right), \quad t \geq s+1, \tag{26}$$

*with $S_s > 1$. Then $\forall t \geq s$,*

$$\ln^3(S_t) \leq \ln^3(S_s) + C_2(s)(t-s), \tag{27}$$

*where $C_2(s) := \frac{3\gamma^2}{2} + \frac{3\gamma^4}{4\ln^3(S_s)} + \frac{\gamma^6}{8\ln^6(S_s)}$.*

*Consequently, for all $t \geq s$,*

$$S_t \leq \exp\left(\left(\ln^3(S_s) + C_3(s)(t-s)\right)^{1/3}\right). \tag{28}$$

*Proof.* From equation 26, for all $k \geq s+1$,

$$\ln(S_k) - \ln(S_{k-1}) = \ln\left(1 + \frac{\gamma^2}{2\ln^2(S_{k-1})}\right). \tag{29}$$

Using $\ln(1+x) \leq x$, $\forall x \geq 0$ and defining $\Delta_k := \ln(S_k) - \ln(S_{k-1})$, we obtain

$$\Delta_k \leq \frac{\gamma^2}{2\ln^2(S_{k-1})}. \tag{30}$$

Now,

$$\begin{aligned}
\ln^3(S_k) - \ln^3(S_{k-1}) &= (\ln(S_{k-1}) + \Delta_k)^3 - \ln^3(S_{k-1}) \\
&= 3\ln^2(S_{k-1})\Delta_k + 3\ln(S_{k-1})\Delta_k^2 + \Delta_k^3.
\end{aligned} \tag{31}$$

We bound the three terms on the RHS using equation 30:-
First term:

$$3\ln^2(S_{k-1})\Delta_k \leq \frac{3\gamma^2}{2}. \tag{32}$$

Second term:

$$3\ln(S_{k-1})\Delta_k^2 \leq 3\ln(S_{k-1})\left(\frac{\gamma^2}{2\ln^2(S_{k-1})}\right)^2 = \frac{3\gamma^4}{4\ln^3(S_{k-1})}. \tag{33}$$

Third term:

$$\Delta_k^3 \leq \left(\frac{\gamma^2}{2\ln^2(S_{k-1})}\right)^3 = \frac{\gamma^6}{8\ln^6(S_{k-1})}. \tag{34}$$

Using these in equation 31, for all $k \geq s+1$,

$$\begin{aligned}
\ln^3(S_k) - \ln^3(S_{k-1}) &\leq \frac{3\gamma^2}{2} + \frac{3\gamma^4}{4\ln^3(S_{k-1})} + \frac{\gamma^6}{8\ln^6(S_{k-1})} \\
&\overset{(1)}{\leq} \frac{3\gamma^2}{2} + \frac{3\gamma^4}{4\ln^3(S_s)} + \frac{\gamma^6}{8\ln^6(S_s)} \\
&:= C_2(s).
\end{aligned} \tag{35}$$

(1) uses that since $(S_t)$ is increasing, $(\ln(S_t))$ is increasing and hence for all $k \geq s+1$,

$$\ln(S_{k-1}) \geq \ln(S_s) > 0.$$

Summing from $k = s+1$ to $t$ yields for all $t \geq s$,

$$\ln^3(S_t) = \ln^3(S_s) + \sum_{k=s+1}^{t} \left(\ln^3(S_k) - \ln^3(S_{k-1})\right) \leq \ln^3(S_s) + (t-s)C_2(s). \tag{36}$$

The upper bound on $S_t$ follows by exponentiating. □

**Lemma A.3.** *Define*

$$\tau_1 := \inf\left\{t \geq 0 : \ln(S_t) > -\sqrt{2F(\mathbf{w}_0)}\right\}.$$

*Then for every integer $t \leq \tau_1$ we have*

$$\ln^2(S_{t-1}) \geq 2F(\mathbf{w}_0),$$

*and consequently*

$$S_t = S_{t-1}\left(1 + \frac{\gamma^2}{2\ln^2(S_{t-1})}\right). \tag{37}$$

*Define*

$$a := 1 + \frac{\gamma^2}{2\ln^2(S_0)}, \qquad b := 1 + \frac{\gamma^2}{4F(\mathbf{w}_0)}.$$

*Then for all integers $t \leq \tau_1$ the sequence satisfies the one-step bounds*

$$S_{t-1}a \leq S_t \leq S_{t-1}b, \tag{38}$$

*and hence the geometric growth rate*

$$S_0 a^t \leq S_t \leq S_0 b^t. \tag{39}$$

*Moreover, $\tau_1$ obeys*

$$\frac{-\sqrt{2F(\mathbf{w}_0)} - \ln(S_0)}{\ln(b)} < \tau_1 \leq 1 + \frac{-\sqrt{2F(\mathbf{w}_0)} - \ln(S_0)}{\ln(a)} \tag{40}$$

*and $\ln(S_{\tau_1})$ satisfies*

$$-\sqrt{2F(\mathbf{w}_0)} < \ln(S_{\tau_1}) \leq -\sqrt{2F(\mathbf{w}_0)} + \ln(b). \tag{41}$$

*Proof.* By definition of $\tau_1$, for all integers $t \leq \tau_1$,

$$\ln(S_{t-1}) \leq -\sqrt{2F(\mathbf{w}_0)} \implies \ln^2(S_{t-1}) \geq 2F(\mathbf{w}_0),$$

so $\max\{2F(\mathbf{w}_0), \ln^2(S_{t-1})\} = \ln^2(S_{t-1})$ and equation 37 follows.

From equation 37, the multiplicative factor is $> 1$, hence $(S_t)$ is nondecreasing and $S_{t-1} \geq S_0$. Since $S_0 \leq S_{t-1} \leq e^{-\sqrt{2F(\mathbf{w}_0)}} < 1$ for all $t \leq \tau_1$, the function $\ln(\cdot)$ is increasing and negative on this range, giving

$$\ln(S_{t-1}) \geq \ln(S_0) \implies \ln^2(S_{t-1}) \leq \ln^2(S_0).$$

Therefore,

$$\frac{\gamma^2}{2\ln^2(S_{t-1})} \geq \frac{\gamma^2}{2\ln^2(S_0)},$$

yielding the lower bound in equation 38. Also, $\ln^2(S_{t-1}) \geq 2F(\mathbf{w}_0)$ implies

$$\frac{\gamma^2}{2\ln^2(S_{t-1})} \leq \frac{\gamma^2}{4F(\mathbf{w}_0)},$$

giving the upper bound in equation 38.
Iterating equation 38 yields equation 39.

If $S_0 b^t < e^{-\sqrt{2F(\mathbf{w}_0)}}$, then by equation 39, $S_t \leq S_0 b^t < e^{-\sqrt{2F(\mathbf{w}_0)}}$, i.e. $\ln(S_t) < -\sqrt{2F(\mathbf{w}_0)}$, so the crossing has not yet occurred and $\tau_1 > t$. This implies the lower bound in equation 40. We have $S_0 a^{\tau_1 - 1} \leq S_{\tau_1 - 1} \leq -\sqrt{2F(\mathbf{w}_0)}$. Using this we get the upper bound in equation 40.

By definition of $\tau_1$, $\ln(S_{\tau_1}) > -\sqrt{2F(\mathbf{w}_0)}$, hence $S_{\tau_1} > e^{-\sqrt{2F(\mathbf{w}_0)}}$, which gives the left inequality in equation 41. Since $\tau_1$ is the first time the inequality $\ln(S_t) > -\sqrt{2F(\mathbf{w}_0)}$ holds, we have $\ln(S_{\tau_1 - 1}) \leq -\sqrt{2F(\mathbf{w}_0)}$, i.e. $S_{\tau_1 - 1} \leq e^{-\sqrt{2F(\mathbf{w}_0)}}$. Applying the one-step upper bound in equation 38 at $t = \tau_1$ yields

$$S_{\tau_1} \leq S_{\tau_1 - 1} b \leq e^{-\sqrt{2F(\mathbf{w}_0)}} b,$$

which proves the right inequality in equation 41. $\qquad\square$

**Lemma A.4.** *Define* $\tau_2 := \inf\{t \geq 0 : \ln(S_t) > \sqrt{2F(\mathbf{w}_0)}\}$.

*Let* $b := 1 + \frac{\gamma^2}{4F(\mathbf{w}_0)}$.

$$\tau_2 < 1 + \tau_1 + \frac{\sqrt{2F(\mathbf{w}_0)} - \ln(S_{\tau_1})}{\ln(b)}. \tag{42}$$

$$\tau_2 \geq \tau_1 + \frac{\sqrt{2F(\mathbf{w}_0)} - \ln(S_{\tau_1})}{\ln(b)}. \tag{43}$$

*Moreover,* $\sqrt{2F(\mathbf{w}_0)} \leq \ln(S_{\tau_2}) \leq \sqrt{2F(\mathbf{w}_0)} + \ln(b)$. *and* $S_t = S_0 b^t$, *for every* $0 \leq 1 \leq \tau_2$.

*Proof.* By definition of $\tau_2$, we have the two boundary conditions

$$\ln(S_{\tau_2 - 1}) \leq \sqrt{2F(\mathbf{w}_0)} \text{ and } \ln(S_{\tau_2}) > \sqrt{2F(\mathbf{w}_0)}. \tag{44}$$

For $\tau_1 < t \leq \tau_2$, we are in the phase where $\max\{2F(\mathbf{w}_0), \ln^2(S_{t-1})\} = 2F(\mathbf{w}_0)$. Hence,

$$\eta_t = \frac{S_{t-1}}{4F(\mathbf{w}_0)} \implies S_t = S_{t-1} + \gamma^2 \eta_t = S_{t-1}\left(1 + \frac{\gamma^2}{4F(\mathbf{w}_0)}\right). \tag{45}$$

Iterating the recurrence gives, for every $\tau_1 \leq t \leq \tau_2$,

$$S_t = S_{\tau_1} b^{t - \tau_1}, \tag{46}$$

Combining equation 44 with equation 46 at $t = \tau_2$ :

$$\ln(S_{\tau_2}) = \ln(S_{\tau_1} b^{\tau_2 - \tau_1}) = \ln(S_{\tau_1}) + (\tau_2 - \tau_1)\ln(b) > \sqrt{2F(\mathbf{w}_0)}.$$

$$\ln(S_{\tau_2 - 1}) = \ln(S_{\tau_1} b^{\tau_2 - \tau_1 - 1}) = \ln(S_{\tau_1}) + (\tau_2 - \tau_1 - 1)\ln(b) \leq \sqrt{2F(\mathbf{w}_0)}.$$

$$\tau_2 \geq \tau_1 + \frac{\sqrt{2F(\mathbf{w}_0)} - \ln(S_{\tau_1})}{\ln(b)}. \tag{47}$$

$$\tau_2 < \tau_1 + 1 + \frac{\sqrt{2F(\mathbf{w}_0)} - \ln(S_{\tau_1})}{\ln(b)}. \tag{48}$$

Next,

$$\ln(S_{\tau_2}) = \ln(S_{\tau_2 - 1}) + \ln(b) \leq \sqrt{2F(\mathbf{w}_0)} + \ln(b),$$

where we used equation 44. Together with $\ln(S_{\tau_2}) \geq \sqrt{2F(\mathbf{w}_0)}$ from equation 44, we obtain

$$\sqrt{2F(\mathbf{w}_0)} < \ln(S_{\tau_2}) \leq \sqrt{2F(\mathbf{w}_0)} + \ln(b). \tag{49}$$

$\qquad\square$

**Theorem 3.3.** *Consider the GD update rule equation 6 with the proposed step-size for logistic regression and consider initialization $\mathbf{w}_0 = \mathbf{0}$. Under Assumption (3.1), we prove the following*

1. *$\forall t \geq 0$, $\mathcal{L}(\mathbf{w}_t) \leq \frac{1}{\eta_t}$ ensuring monotonically decreasing loss iterates.*

2. *There exist constants $c, C > 0$ depending only on $\gamma$ and the initialization such that $\forall t \geq 1$,*

$$\mathcal{L}(\mathbf{w}_t) \leq \frac{Ct^{2/3}}{\exp(ct^{1/3})} = \exp(-\Omega(t^{1/3})). \tag{9}$$

*Proof.* We prove the first part by strong induction on $t$.
*Base case* $(t = 0)$. With the choice of $\eta_0$ and $\mathcal{L}(\mathbf{w}_0) \leq \ln(1 + \exp(\|\mathbf{w}_0\|))$, we get $\mathcal{L}(\mathbf{w}_0)\eta_0 \leq 1$.
*Induction step.* Fix any $t \geq 1$ and assume that for all $k \in \{1, \ldots, t-1\}$,

$$\mathcal{L}(\mathbf{w}_k) \leq \frac{1}{\eta_k}.$$

Then the premise of Lemma (3.2) holds with $s = 0$ and with $F(\mathbf{w}) := \frac{1}{n}\sum_{i=1}^{n}\exp(-y_i\mathbf{x}_i^\top\mathbf{w})$, it gives the following bound,

$$\mathcal{L}(\mathbf{w}_t) \leq \frac{2F(\mathbf{w}_0) + \ln^2(S_{t-1}))}{S_{t-1}} \leq \frac{2\max\left\{2F(\mathbf{w}_0),\ \ln^2(S_{t-1})\right\}}{S_{t-1}} \overset{(1)}{\leq} \frac{1}{\eta_t}, \tag{50}$$

where (1) follows from our step-size scheme. This completes the proof for the first part using induction. With this property, equation 8 gives that the loss values are monotonically decreasing.

Now we prove the second claim.

As $\mathbf{w}_0 = \mathbf{0} \implies \ln(S_{\tau_2}) \geq \sqrt{2F(\mathbf{w}_0)} \geq 1$ (From Lemma A.4).
For $t \geq 1$, the recurrence relation gives the following.

$$\begin{aligned} S_t &= S_{t-1} + \gamma^2\eta_t \\ &= S_{t-1}\left(1 + \frac{\gamma^2}{2\max\left\{2F(\mathbf{w}_0),\ \ln^2(S_{t-1})\right\}}\right), \end{aligned} \tag{51}$$

where the last equality uses the proposed step-size equation 16.

In particular, $(S_t)_{t\geq0}$ is increasing and $S_t \geq S_0,\ \forall t \geq 0$.
Recall definition of $\tau_1, \tau_2, a, b$.

$$\tau_1 := \inf\{t \geq 0 : \ln(S_t) > -\sqrt{2F(\mathbf{w}_0)}\}.$$

$$\tau_2 := \inf\{t \geq 0 : \ln(S_t) > \sqrt{2F(\mathbf{w}_0)}\}.$$

$$a := 1 + \frac{\gamma^2}{2\ln^2(S_0)}, \qquad b := 1 + \frac{\gamma^2}{4F(w_0)}.$$

**Case 1 : $t \leq \tau_1$**
For all $t \leq \tau_1$, from equation 39, we have

$$\mathcal{L}(\mathbf{w}_t) \leq \frac{2\ln^2(S_{t-1})}{S_{t-1}} \leq \frac{2\ln^2(S_0)}{S_0 a^{t-1}}.$$

Hence, the claim is true for $t \leq \tau_1$.

**Case 2:** $\tau_1 < t \leq \tau_2$

For $\tau_1 < t \leq \tau_2$, $\max \left\{ 2F(\mathbf{w}_0), \ln^2(S_{t-1}) \right\} = 2F(\mathbf{w}_0)$. Therefore, equation 51 simplifies as

$$S_t = S_{t-1} + \gamma^2 \eta_t = S_{t-1} \left( 1 + \frac{\gamma^2}{4F(\mathbf{w}_0)} \right) = S_{\tau_1} \left( 1 + \frac{\gamma^2}{4F(\mathbf{w}_0)} \right)^{t-\tau_1}. \tag{52}$$

$$\mathcal{L}(\mathbf{w}_t) \leq \frac{4F(\mathbf{w}_0)}{S_{t-1}} = \frac{4F(\mathbf{w}_0)}{S_{\tau_1} b^{t-1-\tau_2}} < \frac{4b^{\tau_2+1}}{e^{-\sqrt{2}} b^t}.$$

Hence, the claim is true for $\tau_1 < t \leq \tau_2$.

**Case 3:** $t > \tau_2$

Since $\mathbf{w}_0 = \mathbf{0}$, therefore, $\ln(S_{\tau_2}) > \sqrt{2F(\mathbf{w}_0)} \geq 1$. Hence for $t > \tau_2$ the premise of Lemma (A.1) and Lemma (A.2) is satisfied for $s = \tau_2$. For $t > \tau_2$, we have

$$\mathcal{L}(\mathbf{w}_t) \leq \frac{2 \ln^2(S_{t-1})}{S_{t-1}}.$$

The upper bound on the loss is given as follows.

$$\mathcal{L}(\mathbf{w}_t) \; \leq \; 2 \frac{\ln^2(S_{t-1})}{S_{t-1}} \; \overset{\text{Lemma (A.1), (A.2)}}{\leq} \; 2 \frac{\left( \ln^3(S_{\tau_2}) + C_2(\tau_2)\, (t - \tau_2 - 1) \right)^{\frac{2}{3}}}{\exp\left( \ln^3(S_{\tau_2}) + \frac{3\gamma^2}{2C_1(\tau_2)} (t - \tau_2 - 1) \right)^{\frac{1}{3}}}. \tag{53}$$

From Lemma (A.4), we know that $\tau_2$ depends only on initialization and $\gamma$. Hence, we can replace $\tau_2$ by constants that depend on initialization and $\gamma$.

We have $F(\mathbf{w}_0) = 1$, $\ln(S_0) < 0$ and $b = \ln(1 + \frac{\gamma^2}{4})$, We get the following bounds on $\tau, S_\tau, C_1(\tau), C_2(\tau)$.

Using $\frac{x}{x+1} \leq \ln(1+x) \leq x, \forall x \geq 0$, we get

$$\frac{\gamma^2}{4 + \gamma^2} \; \leq \; \ln\left( 1 + \frac{\gamma^2}{4} \right) = \ln(b) \; \leq \; \frac{\gamma^2}{4}. \tag{54}$$

$$\frac{\gamma^2}{2 \ln^2(S_0) + \gamma^2} \; \leq \; \ln\left( 1 + \frac{\gamma^2}{2 \ln^2(S_0)} \right) = \ln(a) \; \leq \; \frac{\gamma^2}{2 \ln^2(S_0)}. \tag{55}$$

$$\sqrt{2} = \sqrt{2F(\mathbf{w}_0)} < \ln(S_{\tau_2}) \leq \sqrt{2F(\mathbf{w}_0)} + \ln\left( 1 + \frac{\gamma^2}{4} \right) \leq \sqrt{2} + \ln\left( \frac{5}{4} \right) < 1.65. \tag{56}$$

Since $\ln(S_0) < 0$ and $\ln(S_{\tau_1}) > 0$, by direct substitution of equation 55 and equation 54 in the upper and lower bounds of $\tau_1, \tau_2$ obtained from Lemma (A.4) and Lemma (A.3), we obtain,

$$1 + \frac{4}{\gamma^2} \left( \sqrt{2} - \ln(S_0) \right) \leq \tau_2 < 2 + \frac{\gamma^2 + 2 \ln^2(S_0)}{\gamma^2} \left( \sqrt{2} - \ln(S_0) \right) \tag{57}$$

Now using bounds on $S_{\tau_2}$,

$$1 + \frac{\gamma^2}{2(1.65)^2} \leq C_1(\tau_2) = 1 + \frac{\gamma^2}{2 \ln^2(S_{\tau_2})} \leq 1 + \frac{\gamma^2}{4}. \tag{58}$$

$$\frac{3\gamma^2}{2} \leq C_2(\tau_2) \leq \frac{9\gamma^2}{2}. \tag{59}$$

With the above inequalites, we upper bound and lower bound $\ln(S_t)$.

**Lower bound:**

From Lemma (A.1)

$$\ln^3(S_t) > \ln^3(S_{\tau_2}) + \frac{3\gamma^2}{2C_1(\tau_2)}(t - \tau_2) \tag{60}$$

$$= \ln^3(S_\tau) + \frac{3\gamma^2}{2C_1(\tau_2)}t - \frac{3\gamma^2}{2C_1(\tau_2)}\tau_2. \tag{61}$$

Since $\ln(S_{\tau_2}) > \sqrt{2}$, it implies

$$\ln^3(S_{\tau_2}) > 2\sqrt{2}. \tag{62}$$

Moreover, using $1 < C_1(\tau_2) < 2$ gives,

$$\frac{3\gamma^2}{2C_1(\tau_2)}t \geq \frac{3\gamma^2}{4}t, \qquad -\frac{3\gamma^2}{2C_1(\tau_2)}\tau_2 \geq -\frac{3\gamma^2}{2}\tau_2.$$

Hence,

$$\ln^3(S_t) \geq 2\sqrt{2} + \frac{3\gamma^2}{4}t - \frac{3\gamma^2}{2}\tau_2. \tag{63}$$

Now substitute the upper bound on $\tau$ from equation 57,

$$\tau_2 < 2 + \frac{\gamma^2 + 2\ln^2(S_0)}{\gamma^2}\left(\sqrt{2} - \ln(S_0)\right),$$

to eliminate $\tau$:

$$\ln^3(S_t) \geq 2\sqrt{2} + \frac{3\gamma^2}{4}t - \frac{3\gamma^2}{2}\left(2 + \frac{\gamma^2 + 2\ln^2(S_0)}{\gamma^2}\left(\sqrt{2} - \ln(S_0)\right)\right)$$

$$\geq \frac{3\gamma^2}{4}t - 3 + 3\ln^3(S_0) + \frac{3\gamma^2\ln(S_0)}{2} - 3\sqrt{2}\ln^2(S_0) - \frac{1}{\sqrt{2}}. \tag{64}$$

**Upper Bound:**

From Lemma (A.2)

$$\ln^3(S_t) \leq \ln^3(S_{\tau_2}) + C_2(\tau_2)\,t - C_2(\tau_2)\,\tau_2 \qquad \text{(From Lemma (A.2))}$$

$$\leq 5 + \frac{9\gamma^2}{2}t - \frac{3\gamma^2}{2}\tau \qquad \text{(From equation 59)}$$

$$\leq 5 + \frac{9\gamma^2}{2}t - \frac{3\gamma^2}{2}\left(1 + \frac{4}{\gamma^2}\left(\sqrt{2} - \ln(S_0)\right)\right) \qquad \text{(From equation 57)}$$

$$= 5 + \frac{9\gamma^2}{2}t + 6\ln(S_0) - \frac{3\gamma^2}{2} - 6\sqrt{2}. \tag{65}$$

Since $\ln(S_0)$ is negative we may drop it to get the upper bound:

$$\ln^3(S_t) \leq \frac{9\gamma^2}{2}t. \tag{66}$$

Therefore, equation 53, equation 66 and equation 64 yields,

$$\mathcal{L}(\mathbf{w}_t) \leq 2\frac{\ln^2(S_{t-1})}{\ln(S_{t-1})} \leq \frac{C\,t^{2/3}}{\exp(c\,t^{1/3})},$$

with,

$$c = \left(\frac{3\gamma^2}{4}\right)^{\frac{1}{3}}, \qquad C = \frac{\left(\frac{9\gamma^2}{2}\right)^{\frac{2}{3}}}{\exp(-3 + 3\ln^3(S_0) + \frac{3\gamma^2\ln(S_0)}{2} - 3\sqrt{2}\ln^2(S_0) - \frac{1}{\sqrt{2}})^{\frac{1}{3}}}.$$

This completes the proof. $\qquad\square$

## A.2 Analysis with Stochastic Gradient Descent

We recall the notations from Section (3.3). The SGD update rule is given as

$$\mathbf{w}_0 = \mathbf{0}$$
$$\mathbf{w}_{t+1} = \mathbf{w}_t - \eta_t \nabla \mathcal{L}_{i_t}(\mathbf{w}_t), \ t \geq 0, \tag{67}$$

where $\nabla \mathcal{L}_{i_t}(\mathbf{w}_t) := \nabla \ln\left(1 + \exp\left(-y_{i_t}\mathbf{x}_{i_t}^\top \mathbf{w}_t\right)\right)$ for index $i_t \in [1, n]$ chosen uniformly at random. We propose an adaptive step-size scheme defined as

$$\eta_t := \min\left\{\frac{1}{\varepsilon}, \ \frac{1}{\mathcal{L}_{i_t}(\mathbf{w}_t)}\right\}, \ t \geq 0, \tag{68}$$

with $\varepsilon$ as the optimality gap.

We first state and prove a key lemma formalizes a key probabilistic property used in the drift analysis of the SGD trajectory. Even when conditioning on the event that the algorithm has not yet reached the target accuracy, the sampling distribution of the stochastic index remains uniform. This holds because the pre-hitting event is measurable with respect to the past filtration and therefore does not bias the independent sampling process.

**Lemma A.5.** *For every $t \geq 0$ and every $j \in \{1, \ldots, n\}$,*

$$\mathbb{P}\big(i_t = j \mid \mathcal{F}_t, \ t < \tau\big) = \frac{1}{n} \qquad \textit{almost surely on } \{t < \tau\}.$$

*Proof.* First note $\{t < \tau\} \in \mathcal{F}_t$ because $\{\tau \leq t\} \in \mathcal{F}_t$ and $\{t < \tau\} = \{\tau \leq t\}^c$. Fix any $A \in \mathcal{F}_t$. Since $i_t$ is independent of $\mathcal{F}_t$ and uniform on $\{1, \ldots, n\}$,

$$\mathbb{P}\big(\{i_t = j\} \cap A \cap \{t < \tau\}\big) = \mathbb{P}(i_t = j)\,\mathbb{P}\big(A \cap \{t < \tau\}\big) = \frac{1}{n}\mathbb{P}\big(A \cap \{t < \tau\}\big).$$

This is exactly the defining characterization of the conditional probability $\mathbb{P}(i_t = j \mid \mathcal{F}_t, \{t < \tau\}) = 1/n$ on $\{t < \tau\}$. $\qquad\square$

The next lemma guarantees that whenever the stopping condition has not yet been reached, there must exist at least one datapoint whose loss remains large. Since the sampling rule selects indices uniformly, this implies that with probability at least $1/n$ the algorithm samples a large-loss example at the next iteration. This observation provides the key lower bound on progress used in the drift analysis.

**Lemma A.6.** *On the event $\{t < \tau\}$ there exists a $\mathcal{F}_t$-measurable index $j^\star = j^\star(\omega) \in \{1, \ldots, n\}$ such that $\mathcal{L}_{j^\star}(\mathbf{w}_t) \geq \varepsilon$, and therefore*

$$\mathbb{P}\big(\mathcal{L}_{i_t}(\mathbf{w}_t) \geq \varepsilon \mid \mathcal{F}_t, \ \{t < \tau\}\big) \geq \frac{1}{n} \qquad \textit{a.s. on } \{t < \tau\}.$$

*Proof.* On $\{t < \tau\}$, by definition of $\tau$ we have $\mathcal{L}(\mathbf{w}_t) > \varepsilon$, i.e. $\frac{1}{n}\sum_{i=1}^n \mathcal{L}_i(\mathbf{w}_t) > \varepsilon$. Since each $\mathcal{L}_i(\mathbf{w}_t) \geq 0$, at least one index satisfies $\mathcal{L}_i(\mathbf{w}_t) \geq \varepsilon$. Define

$$j^\star := \min\{\, i \in \{1, \ldots, n\} : \mathcal{L}_i(\mathbf{w}_t) \geq \varepsilon \,\} \quad \text{on } \{t < \tau\}, \qquad j^\star := 1 \text{ on } \{t \geq \tau\}.$$

Because $\mathbf{w}_t$ is $\mathcal{F}_t$-measurable and each $\mathcal{L}_i$ is deterministic, each event $\{\mathcal{L}_i(\mathbf{w}_t) \geq \varepsilon\}$ lies in $\mathcal{F}_t$, so $j^\star$ is $\mathcal{F}_t$-measurable.

On $\{t < \tau\}$ we have $\{i_t = j^\star\} \subseteq \{\mathcal{L}_{i_t}(\mathbf{w}_t) \geq \varepsilon\}$, hence,

$$\mathbb{P}\big(\mathcal{L}_{i_t}(\mathbf{w}_t) \geq \varepsilon \mid \mathcal{F}_t, \ \{t < \tau\}\big) \geq \mathbb{P}\big(i_t = j^\star \mid \mathcal{F}_t, \ \{t < \tau\}\big).$$

Since $j^\star$ is $\mathcal{F}_t$-measurable, Lemma (A.5) gives $\mathbb{P}(i_t = j^\star \mid \mathcal{F}_t, \{t < \tau\}) = 1/n$ on $\{t < \tau\}$. $\qquad\square$

We use the following stopping-time telescoping lemma to convert the drift inequality into an expected hitting-time bound.

**Lemma A.7.** *Let $D_t$ be a nonnegative adapted process and $\tau$ a stopping time. If for some $a > 0$,*

$$\mathbb{E}[D_{t+1} \mid \mathcal{F}_t] \leq D_t - a\,\mathbf{1}\{t < \tau\} \qquad \text{for all } t \geq 0,$$

*then for every integer $T \geq 1$,*

$$\mathbb{E}[D_T] \leq \mathbb{E}[D_0] - a\,\mathbb{E}[\tau \wedge T].$$

*Consequently, $\mathbb{E}[\tau] \leq \mathbb{E}[D_0]/a$.*

*Proof.* Taking total expectations yields $\mathbb{E}[D_{t+1}] \leq \mathbb{E}[D_t] - a\,\mathbb{E}[\mathbf{1}\{t < \tau\}]$. Summing for $t = 0, \dots, T-1$ gives

$$\mathbb{E}[D_T] \leq \mathbb{E}[D_0] - a\sum_{t=0}^{T-1} \mathbb{E}[\mathbf{1}\{t < \tau\}] = \mathbb{E}[D_0] - a\,\mathbb{E}\left[\sum_{t=0}^{T-1} \mathbf{1}\{t < \tau\}\right].$$

For integer-valued $\tau$, $\sum_{t=0}^{T-1} \mathbf{1}\{t < \tau\} = \min\{\tau, T\} = \tau \wedge T$, proving the first claim. Since $D_T \geq 0$, we get $\mathbb{E}[\tau \wedge T] \leq \mathbb{E}[D_0]/a$. Letting $T \to \infty$ and using monotone convergence ($\tau \wedge T \uparrow \tau$) yields $\mathbb{E}[\tau] \leq \mathbb{E}[D_0]/a$. □

Next, we state and prove an auxiliary lemma.

**Lemma A.8.** *Fix any comparator $\mathbf{u} \in \mathbb{R}^d$. For every $t \geq 0$,*

$$\mathbb{E}_t\left[\|\mathbf{w}_{t+1} - \mathbf{u}\|^2\right] \leq \|\mathbf{w}_t - \mathbf{u}\|^2 - \mathbb{E}_t\left[\min\left\{\frac{\mathcal{L}_{i_t}(\mathbf{w}_t)}{\varepsilon}, 1\right\}\right] + \frac{2}{\varepsilon}\mathcal{L}(\mathbf{u}). \tag{69}$$

*Moreover, on the event $\{t < \tau\}$, if $\mathcal{L}(\mathbf{u}) \leq \varepsilon/(4n)$ then*

$$\mathbb{E}_t\left[\|\mathbf{w}_{t+1} - \mathbf{u}\|^2\right] \leq \|\mathbf{w}_t - \mathbf{u}\|^2 - \frac{1}{2n}. \tag{70}$$

*Proof.* From equation 67,

$$\begin{aligned}
\|\mathbf{w}_{t+1} - \mathbf{u}\|^2 &= \|\mathbf{w}_t - \mathbf{u} - \eta_t \nabla\mathcal{L}_{i_t}(\mathbf{w}_t)\|^2 \\
&= \|\mathbf{w}_t - \mathbf{u}\|^2 - 2\eta_t\langle\nabla\mathcal{L}_{i_t}(\mathbf{w}_t), \mathbf{w}_t - \mathbf{u}\rangle + \eta_t^2 \|\nabla\mathcal{L}_{i_t}(\mathbf{w}_t)\|^2.
\end{aligned}$$

By convexity of $\mathcal{L}_{i_t}$, $\mathcal{L}_{i_t}(\mathbf{w}_t) - \mathcal{L}_{i_t}(\mathbf{u}) \leq \langle\nabla\mathcal{L}_{i_t}(\mathbf{w}_t), \mathbf{w}_t - \mathbf{u}\rangle$, hence

$$\|\mathbf{w}_{t+1} - \mathbf{u}\|^2 \leq \|\mathbf{w}_t - \mathbf{u}\|^2 - 2\eta_t\left(\mathcal{L}_{i_t}(\mathbf{w}_t) - \mathcal{L}_{i_t}(\mathbf{u})\right) + \eta_t^2 \|\nabla\mathcal{L}_{i_t}(\mathbf{w}_t)\|^2.$$

For Logistic Regression we have, $\|\nabla\mathcal{L}_{i_t}(\mathbf{w}_t)\|^2 \leq \mathcal{L}_{i_t}(\mathbf{w}_t)^2$, so

$$\|\mathbf{w}_{t+1} - \mathbf{u}\|^2 \leq \|\mathbf{w}_t - \mathbf{u}\|^2 - 2\eta_t\mathcal{L}_{i_t}(\mathbf{w}_t) + 2\eta_t\mathcal{L}_{i_t}(\mathbf{u}) + \eta_t^2\mathcal{L}_{i_t}(\mathbf{w}_t)^2.$$

By definition, $\eta_t \leq 1/\mathcal{L}_{i_t}(\mathbf{w}_t)$, hence $\eta_t\mathcal{L}_{i_t}(\mathbf{w}_t) \leq 1$ and therefore

$$\eta_t^2\mathcal{L}_{i_t}(\mathbf{w}_t)^2 = (\eta_t\mathcal{L}_{i_t}(\mathbf{w}_t))^2 \leq \eta_t\mathcal{L}_{i_t}(\mathbf{w}_t).$$

Substituting gives the pathwise inequality

$$\|\mathbf{w}_{t+1} - \mathbf{u}\|^2 \leq \|\mathbf{w}_t - \mathbf{u}\|^2 - \eta_t\mathcal{L}_{i_t}(\mathbf{w}_t) + 2\eta_t\mathcal{L}_{i_t}(\mathbf{u}).$$

By equation 67,

$$\eta_t\mathcal{L}_{i_t}(\mathbf{w}_t) = \min\left\{\frac{\mathcal{L}_{i_t}(\mathbf{w}_t)}{\varepsilon}, 1\right\}, \qquad 2\eta_t\mathcal{L}_{i_t}(u) \leq \frac{2}{\varepsilon}\mathcal{L}_{i_t}(\mathbf{u}).$$

Taking $\mathbb{E}_t[\cdot]$ and using $\mathbb{E}_t[\mathcal{L}_{i_t}(\mathbf{u})] = \mathcal{L}(u)$ (uniform sampling) yields equation 69. Since $\{t < \tau\} \in \mathcal{F}_t$, the indicator $\mathbf{1}\{t < \tau\}$ is $\mathcal{F}_t$-measurable. Using $\min\{x, 1\} \geq \mathbf{1}\{x \geq 1\}$, we have

$$\min\left\{\frac{\mathcal{L}_{i_t}(\mathbf{w}_t)}{\varepsilon}, 1\right\} \geq \mathbf{1}\{\mathcal{L}_{i_t}(\mathbf{w}_t) \geq \varepsilon\}.$$

Therefore,

$$
\mathbb{E}_t\left[\min\left\{\frac{\mathcal{L}_{i_t}(\mathbf{w}_t)}{\varepsilon}, 1\right\}\right] \geq \mathbb{E}_t[\mathbf{1}\{\mathcal{L}_{i_t}(\mathbf{w}_t) \geq \varepsilon\}]
$$
$$
= \mathbb{E}_t[\mathbf{1}\{t < \tau\}\,\mathbf{1}\{\mathcal{L}_{i_t}(\mathbf{w}_t) \geq \varepsilon\}] + \mathbb{E}_t[\mathbf{1}\{t \geq \tau\}\,\mathbf{1}\{\mathcal{L}_{i_t}(\mathbf{w}_t) \geq \varepsilon\}]
$$
$$
\geq \mathbf{1}\{t < \tau\}\,\mathbb{P}(\mathcal{L}_{i_t}(\mathbf{w}_t) \geq \varepsilon \mid \mathcal{F}_t,\ t < \tau),
$$

where we used $\mathbf{1}\{t < \tau\} \in \mathcal{F}_t$ to pull it out of the conditional expectation and dropped a nonnegative term. By Lemma (A.6),

$$
\mathbb{P}(\mathcal{L}_{i_t}(\mathbf{w}_t) \geq \varepsilon \mid \mathcal{F}_t,\ t < \tau) \geq \frac{1}{n} \quad \text{a.s. on } \{t < \tau\},
$$

hence,

$$
\mathbb{E}_t\left[\min\left\{\frac{\mathcal{L}_{i_t}(\mathbf{w}_t)}{\varepsilon}, 1\right\}\right] \geq \frac{1}{n}\,\mathbf{1}\{t < \tau\}.
$$

Plugging into equation 69 and multiplying by $\mathbf{1}\{t < \tau\}$ on both sides , using $\mathbf{1}\{t+1 < \tau\} \leq \mathbf{1}\{t < \tau\}$ and $\mathcal{L}(\mathbf{u}) \leq \varepsilon/(4n)$ gives,

$$
\mathbb{E}_t\left[\|\mathbf{w}_{t+1} - \mathbf{u}\|^2\,\mathbf{1}\{t+1 < \tau\}\right] \leq \|\mathbf{w}_t - \mathbf{u}\|^2\,\mathbf{1}\{t < \tau\} - \frac{1}{n}\mathbf{1}\{t < \tau\}
$$
$$
+ \frac{2}{\varepsilon} \cdot \frac{\varepsilon}{4n}\mathbf{1}\{t < \tau\} \tag{71}
$$
$$
= \|\mathbf{w}_t - \mathbf{u}\|^2\,\mathbf{1}\{t < \tau\} - \frac{1}{2n}\mathbf{1}\{t < \tau\}.
$$

On $\{t < \tau\}$ this yields equation 70.

$\square$

**Theorem 3.4.** *Under Assumption (3.1), Adaptive SGD equation 10 on logistic regression with the proposed step-size satisfies,*
$$
\mathbb{E}[\tau] \ \leq\ \frac{2n}{\gamma^2}\,\ln^2\!\left(\frac{4n}{\varepsilon}\right).
$$

*Consequently, for any $\delta \in (0, 1)$,*
$$
\mathbb{P}\left(\tau < \frac{1}{\delta} \cdot \frac{2n}{\gamma^2}\,\ln^2\!\left(\frac{4n}{\varepsilon}\right)\right) \geq 1 - \delta.
$$

*Proof.* Choose
$$
c := \frac{1}{\gamma}\ln\!\left(\frac{4n}{\varepsilon}\right), \qquad \mathbf{u} := c\mathbf{w}^\star.
$$

$\mathcal{L}(\mathbf{u}) \leq e^{-\gamma c} = \varepsilon/(4n)$. Define $D_t := \|\mathbf{w}_t - \mathbf{u}\|^2\mathbf{1}\{t < \tau\} \geq 0$. Lemma (A.8) equation (71) yields,

$$
\mathbb{E}[D_{t+1} \mid \mathcal{F}_t] \leq D_t - \frac{1}{2n}\,\mathbf{1}\{t < \tau\}.
$$

Apply Lemma (A.7) with $a = 1/(2n)$ to obtain

$$
\mathbb{E}[\tau] \leq \frac{\mathbb{E}[D_0]}{a} = 2n\,\|\mathbf{w}_0 - \mathbf{u}\|^2 = 2n\,\|\mathbf{u}\|^2.
$$

Since $\|\mathbf{w}^\star\| = 1$, $\|\mathbf{u}\|^2 = c^2 = \gamma^{-2}\ln^2\!\left(\frac{4n}{\varepsilon}\right)$, so

$$
\mathbb{E}[\tau] \leq \frac{2n}{\gamma^2}\,\ln^2\!\left(\frac{4n}{\varepsilon}\right).
$$

Finally, Markov's inequality gives $\mathbb{P}(\tau \geq \mathbb{E}[\tau]/\delta) \leq \delta$, hence $\mathbb{P}(\tau < \mathbb{E}[\tau]/\delta) \geq 1 - \delta$, and substituting the bound on $\mathbb{E}[\tau]$ finishes the proof. $\square$

## A.3 Analysis with Block Adaptive SGD

We briefly recall the setup of Block Adaptive SGD.
Fix $\varepsilon_0 \in (0,1)$ and define $\varepsilon_k = \varepsilon_0/2^k$, $s_0 = 0$, $s_{k+1} = s_k + N_k$ for $N_k > 0$. During block $k$ (iterations $t \in \{s_k, \ldots, s_{k+1} - 1\}$), run

$$\mathbf{w}_{t+1} = w_t - \eta_t \nabla \mathcal{L}_{i_t}(\mathbf{w}_t), \qquad \eta_t = \min\left\{\frac{1}{\varepsilon_k}, \frac{1}{\mathcal{L}_{i_t}(\mathbf{w}_t)}\right\}.$$

where $i_t$ is chosen uniformly at random,
For a target $\varepsilon \in (0, \varepsilon_0]$, define the activated block

$$k_\varepsilon = \min\{k : \ \varepsilon_k \leq \varepsilon\}.$$

**Theorem 3.5.** *Fix $\delta \in (0,1)$, let $\varepsilon_k := \frac{\varepsilon_0}{2^k}$, $N_k := \left\lceil \frac{4n}{\delta\gamma^2} \ln^2\left(\frac{8n}{\delta\varepsilon_k}\right) \right\rceil$. Let $\varepsilon \in (0, \varepsilon_0]$ be a target accuracy and define $k_\varepsilon := \min\{k : \ \varepsilon_k \leq \varepsilon\}$, $\bar{\varepsilon} := \varepsilon_{k_\varepsilon}$. Define the post-activation hitting time $\tau := \inf\{t \geq s_{k_\varepsilon} : \ \mathcal{L}(\mathbf{w}_t) \leq \bar{\varepsilon}\}$.*

*Under Assumption (3.1), Block Adaptive SGD equation 14 applied to logistic regression satisfies:*

   *(i) **(Probability bound)** One has $\mathbb{P}\left(\min_{0 \leq t \leq s_{k_\varepsilon+1}} \mathcal{L}(\mathbf{w}_t) \leq \varepsilon\right) \geq 1 - \delta$.*

   *(ii) **(Expected hitting time inside the activated block)** Let $N := N_{k_\varepsilon}$. Then almost surely,*

$$\mathbb{E}[(\tau - s_{k_\varepsilon})_+ \wedge N] = \mathcal{O}\left(\frac{n}{\gamma^2} \ln^2\left(\frac{n}{\delta\varepsilon}\right)\right).$$

   *(iii) **(Total iteration complexity)** The total number of iterations up to the end of the activated block satisfies*

$$s_{k_\varepsilon+1} = \mathcal{O}\left(\frac{n}{\delta\gamma^2} \ln^3\left(\frac{n}{\delta\varepsilon}\right)\right).$$

*Proof.* Fix the activated block index $k := k_\varepsilon$ and abbreviate

$$\bar{\varepsilon} := \varepsilon_k, \qquad N := N_k, \qquad s := s_k, \qquad s^+ := s_{k+1} = s + N.$$

Define the (post-block-start) hitting time to the *block target*

$$\tau := \inf\{t \geq s : \ \mathcal{L}(\mathbf{w}_t) \leq \bar{\varepsilon}\},$$

and the stopped iterates $\bar{w}_t := \mathbf{w}_{t \wedge \tau}$. Let $\mathcal{F}_t = \sigma(i_0, \ldots, i_{t-1})$. Then $\mathbf{w}_t$ and $\mathcal{L}(\mathbf{w}_t)$ are $\mathcal{F}_t$-measurable, and

$$\{\tau \leq t\} = \{\mathcal{L}(\mathbf{w}_t) \leq \bar{\varepsilon}\} \in \mathcal{F}_t, \qquad \text{so} \qquad \{\tau > t\} \in \mathcal{F}_t.$$

For comparator $\mathbf{u} \in \mathbb{R}^d$. Expanding the update,

$$\|\mathbf{w}_{t+1} - \mathbf{u}\|^2 = \|\mathbf{w}_t - \mathbf{u}\|^2 - 2\eta_t \langle \nabla \mathcal{L}_{i_t}(\mathbf{w}_t), \mathbf{w}_t - \mathbf{u}\rangle + \eta_t^2 \|\nabla \mathcal{L}_{i_t}(\mathbf{w}_t)\|^2.$$

By convexity of $\mathcal{L}_{i_t}$,

$$\langle \nabla \mathcal{L}_{i_t}(\mathbf{w}_t), \mathbf{w}_t - \mathbf{u}\rangle \geq \mathcal{L}_{i_t}(\mathbf{w}_t) - \mathcal{L}_{i_t}(\mathbf{u}),$$

hence

$$\|\mathbf{w}_{t+1} - \mathbf{u}\|^2 \leq \|\mathbf{w}_t - \mathbf{u}\|^2 - 2\eta_t(\mathcal{L}_{i_t}(\mathbf{w}_t) - \mathcal{L}_{i_t}(u)) + \eta_t^2 \|\nabla \mathcal{L}_{i_t}(\mathbf{w}_t)\|^2.$$

We recall the property $\|\nabla \mathcal{L}_{i_t}(\mathbf{w}_t)\| \leq \min\{1, \mathcal{L}_{i_t}(\mathbf{w}_t)\}$ equation 5. Consequently, $\|\nabla \mathcal{L}_{i_t}(\mathbf{w}_t)\|^2 \leq \mathcal{L}_{i_t}(\mathbf{w}_t)^2$. Since $\eta_t \leq 1/\mathcal{L}_{i_t}(\mathbf{w}_t)$ by construction, we have $\eta_t \mathcal{L}_{i_t}(\mathbf{w}_t) \leq 1$ and thus

$$\eta_t^2 \mathcal{L}_{i_t}(\mathbf{w}_t)^2 = (\eta_t \mathcal{L}_{i_t}(\mathbf{w}_t))^2 \leq \eta_t \mathcal{L}_{i_t}(\mathbf{w}_t).$$

Substituting gives the *pathwise* inequality

$$\|\mathbf{w}_{t+1} - \mathbf{u}\|^2 \le \|\mathbf{w}_t - \mathbf{u}\|^2 - \eta_t \mathcal{L}_{i_t}(\mathbf{w}_t) + 2\eta_t \mathcal{L}_{i_t}(\mathbf{u}). \tag{72}$$

During block $k$, $\eta_t = \min\{1/\bar{\varepsilon}, 1/\mathcal{L}_{i_t}(\mathbf{w}_t)\}$, hence

$$\eta_t \mathcal{L}_{i_t}(\mathbf{w}_t) = \min\Big\{\frac{\mathcal{L}_{i_t}(\mathbf{w}_t)}{\bar{\varepsilon}}, 1\Big\}, \qquad 2\eta_t \mathcal{L}_{i_t}(u) \le \frac{2}{\bar{\varepsilon}} \mathcal{L}_{i_t}(u).$$

Taking conditional expectation in equation 72 and using $\mathbb{E}[\mathcal{L}_{i_t}(\mathbf{u}) \mid \mathcal{F}_t] = \mathcal{L}(\mathbf{u})$ yields

$$\mathbb{E}\big[\|\mathbf{w}_{t+1} - \mathbf{u}\|^2 \mid \mathcal{F}_t\big] \le \|\mathbf{w}_t - \mathbf{u}\|^2 - \mathbb{E}\Big[\min\Big\{\frac{\mathcal{L}_{i_t}(\mathbf{w}_t)}{\bar{\varepsilon}}, 1\Big\} \,\Big|\, \mathcal{F}_t\Big] + \frac{2}{\bar{\varepsilon}} \mathcal{L}(\mathbf{u}). \tag{73}$$

Using Lemma (A.6) on $\{\tau > t\}$ we have:

$$\mathbb{P}\big(\mathcal{L}_{i_t}(\mathbf{w}_t) \ge \bar{\varepsilon} \mid \mathcal{F}_t, \ \{\tau > t\}\big) \ge \frac{1}{n} \qquad \text{a.s. on } \{\tau > t\}.$$

Using $\min\{x, 1\} \ge \mathbf{1}\{x \ge 1\}$ gives

$$\mathbb{E}\Big[\min\Big\{\frac{\mathcal{L}_{i_t}(\mathbf{w}_t)}{\bar{\varepsilon}}, 1\Big\} \,\Big|\, \mathcal{F}_t\Big] \ge \frac{1}{n} \mathbf{1}\{\tau > t\}. \tag{74}$$

Define $\bar{\mathbf{w}}_t = \mathbf{w}_{t \wedge \tau}$. On $\{\tau \le t\}$, $\bar{\mathbf{w}}_{t+1} = \bar{\mathbf{w}}_t$; on $\{\tau > t\}$, $\bar{\mathbf{w}}_t = \mathbf{w}_t$ and $\bar{\mathbf{w}}_{t+1} = \mathbf{w}_{t+1}$. Combining equation 73 and equation 74 yields, for all $t \in \{s, \dots, s^+ - 1\}$,

$$\mathbb{E}\big[\|\bar{\mathbf{w}}_{t+1} - \mathbf{u}\|^2 \mid \mathcal{F}_t\big] \le \|\bar{\mathbf{w}}_t - \mathbf{u}\|^2 - \frac{1}{n} \mathbf{1}\{\tau > t\} + \frac{2}{\bar{\varepsilon}} \mathcal{L}(\mathbf{u}) \mathbf{1}\{\tau > t\}. \tag{75}$$

Set

$$\mathbf{u} := \frac{1}{\gamma} \ln\Big(\frac{8n}{\delta\bar{\varepsilon}}\Big) \mathbf{w}^\star.$$

We restate the property equation 4. By the margin assumption, for every $i$,

$$y_i \langle \mathbf{x}_i, \mathbf{u} \rangle = \frac{1}{\gamma} \ln\Big(\frac{8n}{\delta\bar{\varepsilon}}\Big) y_i \langle \mathbf{x}_i, \mathbf{w}^\star \rangle \ge \ln\Big(\frac{8n}{\delta\bar{\varepsilon}}\Big).$$

Therefore

$$\mathcal{L}_i(\mathbf{u}) = \ln\big(1 + \exp(-y_i \langle \mathbf{x}_i, \mathbf{u} \rangle)\big) \le \exp(-y_i \langle \mathbf{x}_i, \mathbf{u} \rangle) \le \frac{\delta\bar{\varepsilon}}{8n}, \qquad \Rightarrow \qquad \mathcal{L}(\mathbf{u}) \le \frac{\delta\bar{\varepsilon}}{8n} < \frac{\bar{\varepsilon}}{4n}.$$

Plugging into equation 75 gives the clean drift

$$\mathbb{E}\big[\|\bar{\mathbf{w}}_{t+1} - \mathbf{u}\|^2 \mid \mathcal{F}_t\big] \le \|\bar{\mathbf{w}}_t - \mathbf{u}\|^2 - \frac{1}{2n} \mathbf{1}\{\tau > t\}, \qquad t = s, \dots, s^+ - 1. \tag{76}$$

Sum equation 76 over $t = s, \dots, s^+ - 1$ and take conditional expectation given $\mathcal{F}_s$. Because $\bar{\mathbf{w}}_s = \mathbf{w}_s$ (since $\tau \ge s$ by definition) and $\|\bar{\mathbf{w}}_{s^+} - \mathbf{u}\|^2 \ge 0$, we get almost surely

$$0 \le \mathbb{E}\big[\|\bar{\mathbf{w}}_{s^+} - \mathbf{u}\|^2 \mid \mathcal{F}_s\big] \le \|\mathbf{w}_s - \mathbf{u}\|^2 - \frac{1}{2n} \mathbb{E}\Big[\sum_{t=s}^{s^+ - 1} \mathbf{1}\{\tau > t\} \,\Big|\, \mathcal{F}_s\Big].$$

Rearranging yields the conditional bound

$$\mathbb{E}\Big[\sum_{t=s}^{s^+ - 1} \mathbf{1}\{\tau > t\} \,\Big|\, \mathcal{F}_s\Big] \le 2n \|\mathbf{w}_s - \mathbf{u}\|^2 \qquad \text{a.s.} \tag{77}$$

The sum is exactly the number of steps in the block that occur strictly before $\tau$, but capped at $N$:

$$\sum_{t=s}^{s^+-1} \mathbf{1}\{\tau > t\} = N \wedge (\tau - s)_+.$$

Thus equation 77 gives

$$\mathbb{E}[(\tau - s)_+ \wedge N \mid \mathcal{F}_s] \leq 2n \|\mathbf{w}_s - \mathbf{u}\|^2 \qquad \text{a.s.} \tag{78}$$

Taking full expectation in equation 77 gives

$$\mathbb{E}\left[\sum_{t=s}^{s^+-1} \mathbf{1}\{\tau > t\}\right] \leq 2n\, \mathbb{E}\|\mathbf{w}_s - \mathbf{u}\|^2.$$

Also,

$$\mathbf{1}\{\tau > s^+\} \leq \frac{1}{N} \sum_{t=s}^{s^+-1} \mathbf{1}\{\tau > t\},$$

because on $\{\tau > s^+\}$ all $N$ indicators are 1, and otherwise the left-hand side is 0. Therefore

$$\mathbb{P}(\tau > s^+) \leq \frac{1}{N} \mathbb{E}\left[\sum_{t=s}^{s^+-1} \mathbf{1}\{\tau > t\}\right] \leq \frac{2n}{N} \mathbb{E}\|\mathbf{w}_s - \mathbf{u}\|^2. \tag{79}$$

We upper bound the growth of $\|\mathbf{w}_t - \mathbf{u}\|^2$ across blocks $0, 1, \ldots, k-1$ using only the *nonexpansive* part of equation 72. For any block $j \leq k-1$ and any $t \in \{s_j, \ldots, s_{j+1}-1\}$, we have $\eta_t \leq 1/\varepsilon_j$, hence from equation 72,

$$\|\mathbf{w}_{t+1} - \mathbf{u}\|^2 \leq \|\mathbf{w}_t - \mathbf{u}\|^2 + 2\eta_t \mathcal{L}_{i_t}(\mathbf{u}) \leq \|\mathbf{w}_t - \mathbf{u}\|^2 + \frac{2}{\varepsilon_j}\mathcal{L}_{i_t}(\mathbf{u}).$$

Taking conditional expectation and using uniform sampling gives

$$\mathbb{E}\left[\|\mathbf{w}_{t+1} - \mathbf{u}\|^2 \mid \mathcal{F}_t\right] \leq \|\mathbf{w}_t - \mathbf{u}\|^2 + \frac{2}{\varepsilon_j}\mathcal{L}(\mathbf{u}).$$

Iterating this inequality over $t = s_j, \ldots, s_{j+1}-1$ yields

$$\mathbb{E}\|\mathbf{w}_{s_{j+1}} - \mathbf{u}\|^2 \leq \mathbb{E}\|\mathbf{w}_{s_j} - \mathbf{u}\|^2 + \frac{2N_j}{\varepsilon_j}\mathcal{L}(\mathbf{u}).$$

Summing over $j = 0, \ldots, k-1$ and using $\mathbf{w}_{s_0} = \mathbf{w}_0 = \mathbf{0}$ (or any fixed initialization; the bound below uses $\|\mathbf{w}_0 - \mathbf{u}\|^2 = \|\mathbf{u}\|^2$ for $\mathbf{w}_0 = 0$) gives

$$\mathbb{E}\|\mathbf{w}_s - \mathbf{u}\|^2 \leq \|\mathbf{u}\|^2 + 2\mathcal{L}(\mathbf{u}) \sum_{j=0}^{k-1} \frac{N_j}{\varepsilon_j}. \tag{80}$$

Because $N_j$ is nondecreasing in $j$ and $\varepsilon_j = \varepsilon_0/2^j$,

$$\sum_{j=0}^{k-1} \frac{N_j}{\varepsilon_j} \leq N \sum_{j=0}^{k-1} \frac{1}{\varepsilon_j} = \frac{N}{\varepsilon_0} \sum_{j=0}^{k-1} 2^j = \frac{N}{\varepsilon_0}(2^k - 1) < \frac{N}{\varepsilon_k} = \frac{N}{\bar{\varepsilon}}.$$

Now, $\mathbf{u} = \gamma^{-1} \ln(8n/\delta\bar{\varepsilon})\mathbf{w}^\star$, and using $\mathcal{L}(\mathbf{u}) \leq \delta\bar{\varepsilon}/(8n)$, we conclude

$$\mathbb{E}\|\mathbf{w}_s - \mathbf{u}\|^2 \leq \|\mathbf{u}\|^2 + 2 \cdot \frac{\delta\bar{\varepsilon}}{8n} \cdot \frac{N}{\bar{\varepsilon}} = \|\mathbf{u}\|^2 + \frac{\delta N}{4n}. \tag{81}$$

Plug equation 81 into equation 79:

$$\mathbb{P}(\tau > s^+) \leq \frac{2n}{N}\|\mathbf{u}\|^2 + \frac{\delta}{2}.$$

By definition of $N = N_k$,

$$N \geq \frac{4n}{\delta\gamma^2}\ln^2\Big(\frac{8n}{\delta\bar{\varepsilon}}\Big) = \frac{4n}{\delta}\|\mathbf{u}\|^2,$$

so $(2n/N)\|\mathbf{u}\|^2 \leq \frac{\delta}{2}$. Hence

$$\mathbb{P}(\tau > s^+) \leq \frac{\delta}{2} + \frac{\delta}{2} = \delta, \qquad \Rightarrow \qquad \mathbb{P}(\tau \leq s^+) \geq 1 - \delta.$$

If $\tau \leq s^+$, then $\min_{t \leq s^+} \mathcal{L}(\mathbf{w}_t) \leq \bar{\varepsilon} \leq \varepsilon$, so

$$\mathbb{P}\Big(\min_{0 \leq t \leq s_{k+1}} \mathcal{L}(\mathbf{w}_t) \leq \varepsilon\Big) \geq 1 - \delta,$$

which proves Theorem (3.5)(i).

Taking expectation in equation 78 and using equation 81 gives

$$\mathbb{E}[(\tau - s)_+ \wedge N] \leq 2n\,\mathbb{E}\|\mathbf{w}_s - \mathbf{u}\|^2 \leq 2n\|\mathbf{u}\|^2 + \frac{\delta N}{2}.$$

This is proves Theorem (3.5)(ii). Since $k = k_\varepsilon$ implies $\bar{\varepsilon} = \varepsilon_k \in (\varepsilon/2, \varepsilon]$, this is $\mathcal{O}\big(\frac{n}{\gamma^2}\ln^2\big(\frac{n}{\delta\varepsilon}\big)\big)$.

By construction, $k_\varepsilon = \min\{k : \varepsilon_0/2^k \leq \varepsilon\} = \Theta(\ln(\varepsilon_0/\varepsilon))$. Moreover $N_j$ is nondecreasing in $j$ and

$$N_j = \Theta\bigg(\frac{n}{\delta\gamma^2}\ln^2\Big(\frac{n}{\delta\varepsilon_j}\Big)\bigg) \leq \Theta\bigg(\frac{n}{\delta\gamma^2}\ln^2\Big(\frac{n}{\delta\varepsilon}\Big)\bigg) \quad \text{for all } j \leq k_\varepsilon,$$

so

$$s_{k_\varepsilon+1} = \sum_{j=0}^{k_\varepsilon} N_j \leq (k_\varepsilon + 1)\,N_{k_\varepsilon} = \mathcal{O}\bigg(\frac{n}{\delta\gamma^2}\ln\Big(\frac{\varepsilon_0}{\varepsilon}\Big)\ln^2\Big(\frac{n}{\delta\varepsilon}\Big)\bigg) = \mathcal{O}\bigg(\frac{n}{\delta\gamma^2}\ln^3\Big(\frac{n}{\delta\varepsilon}\Big)\bigg).$$

This completes the proof. $\qquad\qquad\square$

## B    Results Adapted from Prior Works

**Lemma 3.2.** *(Adapted from Wu et al. (2024, Lemma 11)) Suppose $\mathcal{L}(\mathbf{w}_k) \leq \frac{1}{\eta_k}$ holds $\forall k \in [s, t-1]$ for logistic loss, under Assumption (3.1), we have*

$$\mathcal{L}(\mathbf{w}_t) \leq \frac{2F(\mathbf{w}_s) + \ln^2(\gamma^2 \sum_{k=s}^{t-1} \eta_k)}{\gamma^2 \sum_{k=s}^{t-1} \eta_k},$$

*where $F(\mathbf{w}_s) := \frac{1}{n}\sum_{i=1}^{n}\exp(-y_i \mathbf{x}_i^\top \mathbf{w}_s)$.*

*Proof.* We use the notation $\mathbf{g}_t := \nabla \mathcal{L}(\mathbf{w}_t)$. Take an arbitrary comparator $\mathbf{u}$ which is a scaling of $\mathbf{w}^\star$.

$$\begin{aligned}
\|\mathbf{w}_{t+1} - \mathbf{u}\|^2 &= \|\mathbf{w}_t - \mathbf{u}\|^2 + 2\eta_t \mathbf{g}_t^\top(\mathbf{u} - \mathbf{w}_t) + \eta_t^2\|\mathbf{g}_t\|^2 \\
&\leq \|\mathbf{w}_t - \mathbf{u}\|^2 + 2\eta_t\,(\mathcal{L}(\mathbf{u}) - \mathcal{L}(\mathbf{w}_t)) + \eta_t^2 \mathcal{L}(\mathbf{w}_t) && \text{(Using convexity and } \|\nabla\mathcal{L}(\cdot)\| \leq \mathcal{L}(\cdot)) \\
&\leq \|\mathbf{w}_t - \mathbf{u}\|^2 + 2\eta_t\,(\mathcal{L}(\mathbf{u}) - \mathcal{L}(\mathbf{w}_t)) + \eta_t L(\mathbf{w}_t) && \text{(Under the condition } \mathcal{L}(\mathbf{w}_t) \leq 1/\eta_t) \\
&= \|\mathbf{w}_t - \mathbf{u}\|^2 + 2\eta_t\mathcal{L}(\mathbf{u}) - \eta_t\mathcal{L}(\mathbf{w}_t). && (82)
\end{aligned}$$

Telescoping from $s$ to $t$, we get

$$\|\mathbf{w}_t - \mathbf{u}\|^2 + \sum_{k=s}^{t-1} \eta_k \mathcal{L}(\mathbf{w}_k) \leq \|\mathbf{w}_s - \mathbf{u}\|^2 + 2\sum_{k=s}^{t-1} \eta_k \mathcal{L}(\mathbf{u})$$

$$\implies \sum_{k=s}^{t-1} \eta_k \mathcal{L}(\mathbf{w}_k) \leq 2\mathcal{L}(\mathbf{u}) \sum_{k=s}^{t-1} \eta_k + \|\mathbf{w}_s - \mathbf{u}\|^2. \tag{83}$$

Taking the comparator $\mathbf{u} = (\mathbf{w}_s + \beta\mathbf{w}^\star)$, where $\beta := \dfrac{\ln\left(\gamma^2 \sum_{k=s}^{t-1} \eta_k\right)}{\gamma}$, we have the following bound on $\mathcal{L}(\mathbf{u})$:

$$\mathcal{L}(\mathbf{u}) = \frac{1}{n} \sum_{i=1}^{n} \ln\left(1 + \exp\left(-y_i \mathbf{x}_i^\top \mathbf{w}_s\right) \exp\left(-y_i \mathbf{x}_i^\top \beta\mathbf{w}^\star\right)\right)$$

$$\leq \frac{1}{n} \sum_{i=1}^{n} \exp\left(-y_i \mathbf{x}_i^\top \mathbf{w}_s\right) \exp\left(-y_i \mathbf{x}_i^\top \beta\mathbf{w}^\star\right) \qquad (\text{Using } \ln(1+\theta) \leq \theta \ \forall\theta \geq 0)$$

$$\leq \frac{1}{n} \sum_{i=1}^{n} \exp\left(-y_i \mathbf{x}_i^\top \mathbf{w}_s\right) \exp\left(-\beta\gamma\right) \qquad (\text{Under Assumption (3.1)})$$

$$= \frac{F(\mathbf{w}_s)}{\gamma^2 \sum_{k=s}^{t-1} \eta_k}, \tag{84}$$

where $F(\mathbf{w}_s) := \frac{1}{n} \sum_{i=1}^{n} \exp\left(-y_i \mathbf{x}_i^\top \mathbf{w}_s\right)$.

Furthermore, from equation 8, we have the monotonicity relation $\mathcal{L}(\mathbf{w}_s) \geq \mathcal{L}(\mathbf{w}_{s+1}) \geq \ldots \geq \mathcal{L}(\mathbf{w}_{t-1}) \geq \mathcal{L}(\mathbf{w}_t)$ when $\mathcal{L}(\mathbf{w}_k) \leq \frac{1}{\eta_k} \ \forall k \in [s, t-1]$. Using these in equation 83, we get the following

$$\mathcal{L}(\mathbf{w}_t) \leq \frac{2F(\mathbf{w}_s) + \ln^2\left(\gamma^2 \sum_{k=s}^{t-1} \eta_k\right)}{\gamma^2 \sum_{k=s}^{t-1} \eta_k}. \tag{85}$$

$\square$

## C   Additional Empirical Results

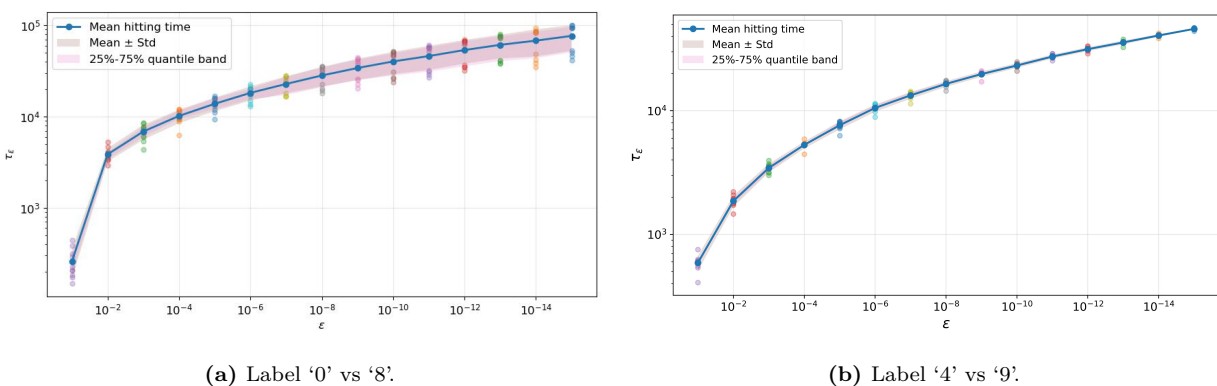

**(a)** Label '0' vs '8'.           **(b)** Label '4' vs '9'.

**Figure 5:** Mean hitting times with our SGD equation (10) for logistic regression over an MNIST subset involving different pairs of labels. The plot shows logarithmic dependence, consistent with the theory.

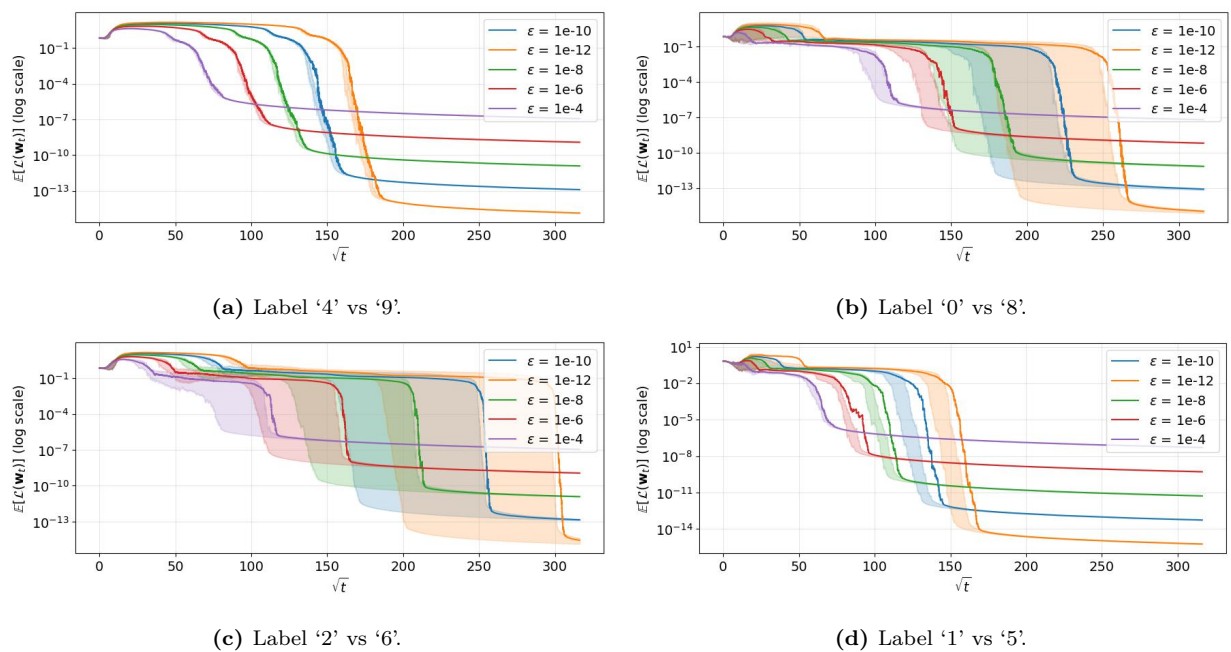

**(a)** Label '4' vs '9'.

**(b)** Label '0' vs '8'.

**(c)** Label '2' vs '6'.

**(d)** Label '1' vs '5'.

**Figure 6:** Average loss values along with 25-75% quantile band with SGD equation 10 for logistic regression over an MNIST subset involving different pair of labels.

