# OpenReview forum: "Stretched Exponential Convergence of (Stochastic) Gradient Descent for Separable Logistic Regression"
_TMLR — Accepted by TMLR_

### Review · Reviewer_TUfS · 2026-04-12

**Summary Of Contributions:**

This is a theoretical paper studying gradient descent (GD) and stochastic gradient descent (SGD) for logistic regression. The paper notes that recent work in this space has demonstrated first order optimization schemes with super-linear convergence rates. However, these schemes either have an initial "unstable" phase, where the iterates may grow in loss, or they have "complex" or adaptive step sizes.

Instead, the authors propose a non-adaptive scheme for GD that (1) achieves exponential convergence, and interestingly, features a *growing* (non-adaptive) step size, which contrasts with the usual requirement of a constant or shrinking step size, and (2) does *not* have an initial unstable phase. The authors provide analogous results for SGD, with the exception that their step-size is adaptive. Empirical results on MNIST and synthetic data show that the theoretical results are not violated in practice.

**Audience:**

Yes

**Audience Explanation:**

The success of large step sizes in modern applied ML is a confusing success story, given classic theory's prescription of using a quickly-decaying step size. I don't believe that this apparent contradiction has been totally resolved, and I think this paper makes a modest step towards shedding more light on this contradiction. And the paper is generally well-written enough to be accessible to an audience beyond the theoretical optimization community -- I am certainly not a member of that community, so I found the paper and its discussion of related work to be educational!

**Claims And Evidence:**

No

**Claims Explanation:**

I think there are a few places where the paper's claims need to be better backed up.

# Claims that the optimization scheme is fully specified in advance.

The paper writes as its main claim about GD: "We propose a new deterministic step-size scheme for GD that is fully specified in advance, prevents loss oscillations, and yields convergence that is any time nearly exponential." I think this is not correct. The GD scheme relies on knowing $\gamma$, which is the constant of separation (i.e., $y_i x_i^T w^* \geq \gamma > 0$ for all $i = 1, \dots, n$). But in practice, as far as I know, we never know $\gamma$. So I actually don't think this scheme is "fully specified in advance" at all, which makes most of the criticisms of previous work from the introduction apply just as well to the proposed method.

# Relationship to Zhang et al. (2025b)
The authors differentiate their work from Zhang et al. (2025b) by noting that Zhang uses an "adaptive or complex learning-rate [schedule]". But Zhang's step size doesn't look overly complex to me; they use:
$$
\eta_t = \eta_0 \frac{\exp (L(w_t))}{\exp(L(w_t)) + 1}.
$$
Just algebraically, this doesn't look any more complex than the scheme proposed in the current paper. I guess one could say that the rule from Zhang et al. (2025b) is "adaptive" in that it depends on the current loss value, whereas the scheme proposed in the current paper doesn't rely on the loss. But what is the significance of this?

# Finiteness of the transition to "fast" convergence
My understanding is that the proofs are meant to show that there are two "stages" to the convergence of the proposed step-size scheme -- one where convergence is polynomial, and then a latter where the convergence is the desired exponential rate. I believe the hitting time $\tau_1$ is supposed to be the time when the GD dynamics transition to their "fast" state. But I didn't see any proof that the hitting time $\tau_1$ needs to be finite. Can the authors comment on whether we should necessarily expect $\tau_1 < \infty$?

My understanding is that this finiteness of $\tau_1$ is essential for the proof of Theorem 3.3 to go through. The paper bounds the loss, $L(w_t)$ for $t \leq \tau_1$ as:
$$
L(w_t) \leq \frac{2 \ln^2(S_0)}{S_0 a^{1-t}},
$$
where $a := 1 + \gamma^2 / (2\ln^2(S_0))$ is a constant. The paper then states that this establishes the desired bound of the theorem for $t \leq \tau_1$ (i.e., that $L(w_t) \leq Ct^{2/3} / \exp(ct^{1/3})$). If $\tau_1$ is finite, then we can just absorb everything into the constant $C$. But otherwise, this bound doesn't seem to hold.

# Experiments aren't providing much
To be clear, the experiments aren't contradicting the theoretical results. But I don't think they're doing much beyond that. I would encourage the authors to think of value of synthetic data (or "psuedo-real" data in the case of MNIST) as being able to illustrate things that are hard to prove theoretically. Currently it's just not clear what readers should be taking away from the experiments other than "an empirical check fails to prove the theory wrong." But there are always many exciting questions that synthetic experiments can answer! As an example, the proofs proceed by doing a lot of lower and upper bounding of things. How loose can these bounds get? How tight can they get?

A small point is that the description of the first SGD experiment technically contradicts itself: "[the experiment] shows exponential convergence ... which illustrates our theoretical claim of near exponential convergence". Is the convergence exponential or near exponential?

**Requested Changes:**

# Critical changes

- Explain how Section 3.3 fits into the paper. I found the transition from Section 3.2 to 3.3 a little confusing. The intro makes it sound like the really cool part of 3.2 is that it doesn't require an adaptive step size. And it makes a big point of why this is important. But now in Section 3.3, we're going back to proposing an adaptive step size. Maybe the results in 3.3 are novel and interesting, but this section really clashed with the setup of the paper.
- Clarify claims that the GD scheme is fully specified in advance.
- Clarify contributions over Zhang et al. (2025b). The standards for TMLR are that well-studied but somewhat incremental papers are acceptable. But I think the difference between Zhang 2025b (and 2025a!) should be more clearly discussed given how similar those papers appear to the current paper.
- Clarify whether we're guaranteed to transition to the "fast" GD dynamics (is $\tau_1$ guaranteed to be finite?)
- Either further explain the value of the current experiments, or replace them with a more detailed empirical investigation, or remove them from the paper.
- Lemma A.2 seems to implicitly assume that there exists an $s$ such that $S_s > 1$. Is this necessarily true? This feels related to the issues of $\tau_1$ being finite.


# Small changes, non-critical
- I would consider whether readers need to read through all of Eqs 2-5; I didn't find them necessary to read the paper.
- "after an oscillating phase of $O(T /2)$" -- this isn't properly defined. What do we mean by "an oscillating phase of [any number here]"? Also, correct use of big-O notation wouldn't include the 1/2 factor.
- "can also induce loss oscillations up to a reasonably long duration of up to $(T /2)$" Is this actually $T/2$? Above the paper said it was $O(T/2)$.
- "This holds because the pre-hitting event,..." this seems like overkill - the sampling scheme is defined to be uniformly at random; how could it possibly be statistically dependent on anything else?

- Sometimes the notation makes things harder to read without offering much convenience for the reader. E.g., writing $\bar \epsilon$ in place of $\epsilon_{k_\epsilon}$ in the statement of Theorem 3.5. or $s$ in place of $s_k$ in the proof sketch of Theorem 3.5. I think this kind of "one-use" notation that doesn't save any space just makes the reader have to keep track of more stuff.

- Near Eq (35), "Inequality (1)" -- is the reference to (1) a typo?
- The notation used in Lemmas A.1 through A.3 is a little confusing. A.1 and A.2 seem to be using an independent definition of $S_t$. But then A.3 seems to be talking about the original definition of $S_t$ used in the main text. It would be good to be more clear about this.

---

> ### Author Response · Authors · 2026-04-22
>
> We thank the reviewer for this clear and accurate summary of our contributions. We appreciate the recognition of both the theoretical aspects of our work and the key distinctions from prior approaches, particularly the absence of an unstable phase and the use of a growing, non-adaptive step size. We are also glad that the empirical validation is viewed as supporting the theoretical findings.
>
> **Q1. "Explain how Section 3.3 fits into the paper. I found the transition from Section 3.2 to 3.3 a little confusing."**
>
> **A1.** We appreciate the opportunity to clarify this. The central objective of this work is to demonstrate that simple algorithmic choices can yield significantly faster convergence rates without requiring a phase-wise analysis. The nature of the step-size choices necessarily differs between GD and SGD due to their underlying dynamics.
>
> In Section (3.2), we show that for deterministic GD, a non-adaptive (yet growing) step-size scheme ensures monotonic decrease of the loss. This monotonicity is crucial for our analysis-specifically, it underpins Lemma (3.2), which is then used to derive the final convergence rate.
>
> However, such monotonic descent is not expected to hold for SGD, even in expectation, due to inherent stochasticity. This necessitates a different analytical approach in Section (3.3). In particular, we introduce a step-size that adapts to the local curvature through the observed stochastic loss, which allows us to control the noise while still obtaining fast convergence.
>
> Importantly, the adaptivity introduced here is minimal and structured: the step sizes depend only on a prescribed tolerance parameter (or block-wise tolerance in Section (3.4)) and the stochastic loss, rather than requiring complex line-search or fully adaptive schemes. Using this construction, Sections (3.3) and (3.4) establish that Adaptive SGD and Block Adaptive SGD achieve stretched exponential convergence rates.
>
> **Q2. "Clarify claims that the GD scheme is fully specified in advance"**
>
> **A2.** We thank the reviewer for raising this point. While the proposed scheme in  (Eq. 7) assumes knowledge of **problem-dependent parameter** $\gamma$, the analysis remains valid when $\gamma$ is replaced by any positive $\gamma_0 \leq \gamma$ in the step size schedule. In particular, the main inductive upper bound (Eq.~50) continues to hold under this substitution. This follows from the monotonicity of the function $\frac{x^2}{\ln^2 x}$ over the relevant range, which ensures that decreasing $\gamma$ preserves the inequality.
>
> $$
> \mathcal{L}(w_t) \le
> \frac{2 \max\left(2F(w_0), \ln^2\Bigl(\sum_{i=1}^{t-1}\gamma^2 \eta_i\Bigr)\right)}
> {\left(\sum_{i=1}^{t-1}\gamma^2 \eta_i\right)}
> \le
> \frac{2 \max\left(2F(w_0), \ln^2\Bigl(\sum_{i=1}^{t-1}\gamma_0^2 \eta_i\Bigr)\right)}
> {\left(\sum_{i=1}^{t-1}\gamma_0^2 \eta_i\right)}.
> $$
>
> This is analogous to classical optimization results, where convergence guarantees remain valid for any step size $\eta \leq \frac{2}{L}$. We also note that the non-adaptive schemes of Wu et al. (2024) require prior knowledge of the optimization horizon $T$, whereas our method does not and their accelerated rate of $O(1/T^2)$ requires choosing $\eta = \frac{\gamma^2T}{120}$, which again depends on $\gamma$.
>
> In practice one may also run a hyperparameter sweep over $\gamma$, which adds only a complexity of $\log(1/\gamma)$ if taking a logarithmic grid for the sweep. We propose to add this in the revised manuscript to clarify the term "fully-specified".
>
> **Q3. "Clarify contributions over Zhang et al. (2025b)."**
>
> **A3.** While we agree that there are connections to Zhang et al. 2025(a, b) in terms of our work analyzing large step-size optimization dynamics and achieving stretched exponential rates, our work is uniquely positioned in several aspects. In particular, Zhang et al. 2025(a, b) prove exponential rates after a burn-in phase of $1/\gamma^2$ by employing adaptive step sizes. Before reaching this burn-in time, they only provide upper bound on the loss at the average iterate and for $t< 1/\gamma^2$, this upper bound is of the form $L(\bar{\mathbf{w}}_t)\le\exp(c\eta/t)$ where $\bar{\mathbf{w}}_t$ is the average iterate at timestep $t$ and $c>0$. Moreover, Zhang et al. 2025(a, b) do not provide any discussion on obtaining exponential rates for SGD. To the best of our knowledge, Wu et al. (2024) is the only work that discusses SGD in the context of large step-size, however, they only provide the unstable phase analysis where they obtain $O(\eta/t)$ upper bound on the average loss. We
> propose a GD scheme with an *increasing non-adaptive step-size* scheme with which the training dynamics never enters unstable phase and the obtained stretched exponential rate of $O(\exp(-\Omega(t^{1/3})))$ is *anytime*. We have accordingly added additional details in the Related Works section.

---

> > ### Author Response · Authors · 2026-04-22
> >
> > **Q4. Clarify whether we're guaranteed to transition to the "fast" GD dynamics (is** $\tau_1$ **guaranteed to be finite?**
> >
> > **A4.** $\tau_1$ is guaranteed to be finite as we prove in the paper. We think the reviewer might have misunderstood certain parts of the proof. There is no inherent ``fast phase'' or slow phase in the behavior of the algorithm; the rates we establish are *anytime guarantees*. That said, for the purpose of analysis, we partition the argument into regimes depending on whether $\ln^2(S_t)$ or $2F(w_0)$ dominates. Asymptotically, $\ln(S_t) \gg \sqrt{2F(w_0)}$ and $\ln^2(S_t) $ is the dominating  term. Before this regime, the sequence exhibits a geometric/exponential growth rate, as characterized by Lemma (A.3) (Eq. 39) and Lemma (A.4) (Eq. 46). The convergence rate is directly tied to growth rate of $S_t$.
> >
> > The reviewer seems to have inadvertently reversed the sign in the exponent of the denominator. For the regime $t \le \tau_1$, the correct bound is
> > $$
> > \mathcal{L}(w_t)\le \frac{2\ln^2(S_0)}{a^{t-1}},$$ whereas the review incorrectly states the denominator as $a^{1-t}$.
> > \
> > However, $\tau_1$ is finite. In particular, Lemma (A.3) (Eq. 41) provides an explicit upper bound on $\tau_1$:
> > $$\tau_1
> > \leq
> > 1+\frac{-\sqrt{2F(w_0)}-\ln(S_0)}{\ln(a)}.$$ To see this directly, note that for any $\gamma > 0$ (linearly separable data), we have
> > $\ln(S_0) = \ln(\eta_0 \gamma^2) > -\infty$.
> > Moreover,
> > $\ln(a) = \ln\left(1 + \frac{\gamma^2}{2 \ln^2(S_0)}\right) > 0$ ,
> > since the argument of the logarithm is strictly greater than $1$. As both the numerator and denominator in the upper bound for $\tau_1$ are finite, therefore $\tau_1$ must be finite.
> >
> > **Q5. Either further explain the value of the current experiments, or replace them with a more detailed empirical investigation, or remove them from the paper.**
> >
> > **A5.** While the primary focus of this work is theoretical, the experiments serve an important role in
> > validating that our analysis captures the correct convergence behavior. In particular, the observed
> > rates are consistent with our theoretical predictions, which is non-trivial and not evident a priori
> > from the analysis alone. This alignment provides evidence that the theory is not only internally
> > sound but also reflective of the empirical optimization dynamics.
> >
> > **Q6. Lemma A.2 seems to implicitly assume that there exists an** $s$ **such that** $S_s > 1$ **. Is this necessarily true? This feels related to the issues of  being finite.**
> >
> > **A6.** Yes, this is tied to the finiteness of $\tau_1$ (which we show in the paper and summarize in A4). In particular, $\tau_1 < \infty$ implies that $\tau_2$ is also finite for $\gamma >0$. Moreover, by the definition of $\tau_2$, we have $\ln(S_{\tau_2}) > \sqrt{2} > 0$.
> >
> > **Q7. I would consider whether readers need to read through all of Eqs 2-5; I didn't find them necessary to read the paper.**
> >
> > **A7.** Equations (2–5) summarize the fundamental properties of the logistic loss that are crucial to our analysis as these properties are utilized in the proofs and largely explains the reasoning for the proposed step-size schemes.
> >
> > **Q8. "What do we mean by "an oscillating phase of [any number here]"? Also, correct use of big-O notation wouldn't include the 1/2 factor.
> > "can also induce loss oscillations up to a reasonably long duration of up to** $T/2$ **" Is this actually** $T/2$**? Above the paper said it was** $O(T/2)$"
> >
> > **A8.** Thank you for pointing this. We have corrected this notational inconsistency. We intended to refer to an oscillating phase up to $T/2$ iterations where $T$ is the time horizon. By oscillating phase, we mean a non-monotonic trajectory of the loss.
> >
> > **Q9. This holds because the pre-hitting event,..." this seems like overkill - the sampling scheme is defined to be uniformly at random; how could it possibly be statistically dependent on anything else?**
> >
> > **A9.** The sampling distribution would become non-uniform when conditioning on future events, for example if we condition on the event that at timestep $t+1$  the loss is less than $\varepsilon$ this would influence what points were sampled at timestep $t$.
> >
> > **Q10. Near Eq (35), "Inequality (1)" -- is the reference to (1) a typo?**
> >
> > **A10.** Thank you for noticing this, we have corrected this typo.
> >
> > **Q11. The notation used in Lemmas A.1 through A.3 is a little confusing. A.1 and A.2 seem to be using an independent definition of** $S_t$**. But then A.3 seems to be talking about the original definition of  used in the main text. It would be good to be more clear about this.**
> >
> > **A11.** All the lemmas (A.1, A.2, A.3) rely on the same underlying definition. In particular, Lemmas (A.1) and (A.2) focus on the regime where $\ln^2(S_t)$ dominates $2F(w_0)$ and $\ln(S_t) > 0$.

---

### Review · Reviewer_V6Gg · 2026-04-12

**Summary Of Contributions:**

This paper studies separable logistic regression under a margin assumption. For full-batch gradient descent (GD), it proposes a deterministic, increasing step-size schedule $\eta_t$ designed to maintain $\mathcal{L}(w_t) \le 1/\eta_t$ for all $t$. The main GD theorem establishes the convergence bound: $L(w_t) \le C t^{2/3}e^{-ct^{1/3}}$.

For stochastic gradient descent (SGD), the paper studies the adaptive rule $\eta_t$, proving an finite expected first-hitting-time bound of $\mathbb{E}[\tau] \le \mathcal{O}(\frac{n}{\gamma^2} \log^2 \frac{n}{\epsilon})$. The authors also propose a block SGD variant intended to remove the target tolerance $\epsilon$, though it comes at the cost of extra factor in total iteration time.

**Audience:**

Yes

**Audience Explanation:**

This work addresses a highly relevant theoretical question within the TMLR optimization community. The main takeaway for GD, that one can obtain faster-than-polynomial decay with a fully predetermined, monotone schedule, is a valuable contribution.

**Broader Impact Concerns:**

I do not have concerns on the ethical implications of the work.

**Claims And Evidence:**

No

**Claims Explanation:**

The paper has several strengths:

- The core idea of the GD improvement is clean. Leveraging the 1-smoothness of $\log \mathcal{L}$ together with a step-size schedule chosen to maintain $\mathcal{L}(w_t) \le 1/\eta_t$ avoids the oscillating phase.
- The proof for the SGD results is well-structured.

However, I’ve also identified several cons for the paper:

- The paper repeatedly mention “exponential convergence”, but the result for GD, for example, is $L(w_t) \le \exp(-\Omega(t^{1/3}))$ which is stretched exponential (weaker than exponential). Similarly for the SGD results. The “exponential convergence” language is somewhat misleading.
- In the contribution, the adaptive SGD method is described as “does not rely on line search and on the final tolerance level.” However, the SGD method discussed in Section 3.3 still has the final tolerance dependence; it is not until Section 3.4 where the block variant is introduced when the final tolerance dependence is removed. However, none of this is discussed in the introduction (or the contributions).
- In the proof, Eq. (79) leads to an unconditional expectation bound for $(\tau - s_k)_+ \wedge N$; I am not entirely sure how that leads directly to the proof of Theorem 3.5 (ii).
- The empirical results for GD compares only to a few constant step-sizes on synthetic data. There is no comparison to the relevant monotone/adaptive baselines mentioned in the Instruction section. The SGD experiment is performed on a small MNIST dataset with no results on the block-adaptive variant. While I understand this is a theory-focused paper, the experiments fail to establish practical significance.

**Requested Changes:**

In addition to the changes needed with respect to the cons described above, several notional/proof issues also need to be fixed.

- In the specification of Theorem 3.3, the statement of "w.l.o.g. consider initialization $w_0 = 0$" is not obvious; it is used later in the proof to bound initial constants.
- In Appendix B, Page 28, the monotonicity relation is written in the wrong direction.
- In the GD proof, the text states "Since $\ln(S_0), \ln(S_{\tau_2}) < 0$" but it is also stated immediately above that $\ln(S_{\tau_2}) > \sqrt{2}$.

---

> ### Author Response · Authors · 2026-04-22
>
> We thank the reviewer for carefully reading the manuscript and for highlighting the key strengths of our work, particularly the clean core idea and a structured analysis that leads to provable improvements.
>
> **Q1. On the terminology ``exponential convergence"**
>
> **A1.** We thank the reviewer for this helpful suggestion on making out contributions more precise. We have accordingly revised the terminology from “exponential convergence” to “stretched exponential convergence”.
>
> **Q2. In the contribution, the adaptive SGD method is described as “does not rely on line search and on the final tolerance level.” However, the SGD method discussed in Section 3.3 still has the final tolerance dependence; it is not until Section 3.4 where the block variant is introduced when the final tolerance dependence is removed. However, none of this is discussed in the introduction (or the contributions).**
>
> **A2.** This is a helpful observation, and we agree that the introduction was not making this distinction sufficiently explicit. We have revised the contributions in the introduction to clearly distinguish these two results and explicitly state that tolerance-independence is achieved via the Block Adaptive extension presented in Section (3.4). This will ensure that the claims are precise and aligned with the technical development.
>
> **Q3. "In the proof, Eq. (79) leads to an unconditional expectation bound for ** $ (\tau-s_{k})_+ \wedge N $; ** I am not entirely sure how that leads directly to the proof of Theorem 3.5 (ii)."**
>
> **A3.** Thank you for pointing this out. Our bound in Theorem 3.5(ii) is in fact on the unconditional expectation. On taking expectations in Equation (78), together with Equation (80), yields the desired result. We have corrected this in the revised manuscript.
>
> **Q4. "There is no comparison to the relevant monotone/adaptive baselines mentioned in the Instruction section."**
>
> **A4.** We appreciate the opportunity to clarify this point. In the context of large step-size optimization, the closest prior works are Wu et al. (2024) and Zhang et al. 2025 (a, b). Wu et al. (2024) establish an $O(1/T^2)$ convergence rate, but only after an initial phase of loss oscillations that can persist for up to $T/2$ iterations. Zhang et al. 2025 (a, b) obtain exponential convergence rates; however, this guarantee holds only after a burn-in period of order $O(1/\gamma^2)$, prior to which they show the loss at the averaged iterate bounded by $O(\exp(\eta/t))$. In contrast, our method achieves an anytime stretched exponential convergence rate of $\exp(-\Omega(t^{1/3}))$, without requiring a burn-in phase and with the loss decreasing monotonically throughout. Moreover, to the best of our knowledge, this is the first approach to establish stretched exponential convergence rates for SGD in separable logistic regression without relying on line-search procedures.\
> Due to these fundamental differences in convergence behavior and assumptions, there are no directly comparable prior work or adaptive baselines in the literature that operate in the same regime as our method.
>
> **Q5. On the performance of Block Adaptive SGD**
>
> **A5.** While Block Adaptive SGD provides a principled way to eliminate the need to know the tolerance level a priori, the theoretical derived block sizes are conservative with dependence of $n$. We found these block sizes are in general larger than what is needed in practice, which obscures its empirical advantage. We have included this observation in Section (5) and motivated improving this aspect in future work.
>
> **Q6. "While I understand this is a theory-focused paper, the experiments fail to establish practical significance".**
>
> **A6.** While the primary focus of this work is theoretical, the experiments serve an important role in validating that our analysis captures the correct convergence behavior. In particular, the observed rates are consistent with our theoretical predictions, which is non-trivial and not evident a priori from the analysis alone. This alignment provides evidence that the theory is not only internally sound but also reflective of the empirical optimization dynamics.
>
> **Q7. In the specification of Theorem 3.3, the statement of "w.l.o.g. consider initialization  $w_0 =0 $" is not obvious; it is used later in the proof to bound initial constants.**
>
> **A7.** Thank you for pointing this. We have removed "w.l.o.g." from the Theorem statement.
>
> **Q8. "In Appendix B, Page 28, the monotonicity relation is written in the wrong direction."**
>
> **A8.** Thank you for sharing this. We have corrected this in the revised manuscript.
>
> **Q9. "In the GD proof, the text states "Since  $\ln(S_0),\ln(S_{\tau_2}) >0$" but it is also stated immediately above that** $\ln(S_{\tau_2}) > \sqrt{2}$."
>
> **A9.** Thank you for pointing this out, $\ln(S_{\tau_2}) < 0$ was a typo. It is actually $\ln(S_{\tau_1})<0$. We have corrected this in the revised manuscript.

---

### Review · Reviewer_anTy · 2026-04-14

**Summary Of Contributions:**

This paper studies the convergence behavior of gradient descent (GD), stochastic gradient descent (SGD) as well as adaptive SGD for separable logistic regression without relying on the edge-of-stability regime. More specifically,

* For GD, the authors proposed a deterministic, non-adaptive and increasing step size schedule to achieve an anytime exponential convergence.

* For SGD, the authors proposed a simple adaptive step size schedule based on the current-step loss and achieved an exponential convergence result that achieves a target tolerance level.

* They further adapted SGD to block adaptive SGD, which removes the dependency on the tolerance level through a doubling-trick strategy an further gave an anytime convergence guarantee.

**Audience:**

Yes

**Audience Explanation:**

These results will be of interest to researchers in optimization and learning theory.

**Claims And Evidence:**

Yes

**Claims Explanation:**

The claims in the paper are clearly supported by Theorems 3.3, 3.4 and 3.5. Each of these theorems are clearly motivated and proof sketches are given to give an overview of the technical steps.

One minor gap: The paper suggests "exponential convergence", but it's really $\exp(-\Omega(t^{1/3}))$ instead of the conventional $\exp(-t)$. The authors should frame this explicitly to avoid overclaiming.

**Requested Changes:**

1. Rephrase the terminology of "exponential convergence" to a more precise statement;

2. The experimental sections could be strengthened. For the SGD experiments, the authors should show both the average loss and the standard deviation across all the training runs. I think it would be interesting to see how the block adaptive SGD performs compared to SGD with predetermined $\epsilon$. There should also be more detailed discussion regarding Figure 3, especially what happens in different phases of the convergence period, and how changing the value of $\epsilon$ alters the behavior of loss change in each phase.

3. Comparison to prior work could be strengthened. I think it would further clarify the contribution if a table consisting of different settings, algorithms and convergence rates comparing this work to prior works is presented. The comparison to prior work could also be conducted in the experiment section.

---

> ### Author Response · Authors · 2026-04-22
>
> We thank the reviewer for the thoughtful feedback. We are encouraged to note that the reviewer found our theorems for GD, SGD and block adaptive SGD clearly
> motivated and appreciated the proof sketches containing overview of the technical steps.
>
> **Q1. On the terminology ``exponential convergence"**
>
> **A1.** We thank the reviewer for this helpful suggestion on making out contributions more precise. We have accordingly revised the terminology from “exponential convergence” to “stretched exponential convergence”.
>
> **Q2. Add standard deviation across all the training runs for SGD experiments**
>
> **A2.** Thank you for the suggestion. We have added plots for our Adaptive SGD that additionally show quantile bands in Appendix Section (C). We have also included plots for the hitting times $\tau_\varepsilon$ for the Adaptive SGD experiment.
>
> **Q3. On the performance of Block Adaptive SGD**
>
> **A3.** While Block Adaptive SGD provides a principled way to eliminate the need to know the tolerance level a priori, the theoretical derived block sizes are conservative with dependence of $n$. We found these block sizes are in general larger than what is needed in practice, which obscures its empirical advantage. We have included this observation and motivated improving this aspect in Section (5) of the future work.
>
> **Q4. "There should also be more
> detailed discussion regarding Figure 3, especially what happens in different phases of the convergence period, and how changing the value of $\varepsilon$ alters the behavior of loss change in each phase"**
>
> **A4.** We thank the reviewer for this constructive feedback. As we discuss in the paper, the SGD dynamics go through a transient phase and then display a sharp descent consistent with the theoretical findings of achieving stretched exponential rate. We further observe that smaller values of $\varepsilon$ delay the onset of this rapid descent but ultimately achieve significantly lower final loss values, whereas larger $\varepsilon$ leads to earlier but less pronounced convergence. This behavior is consistent with our theoretical predictions relating $\varepsilon$ to the optimality gap. We have included this in the revised manuscript.
>
> **Q5. Comparison to prior works**
>
> **A5.**
> While similar in spirit to large step size optimization algorithms, our contribution is uniquely positioned in several aspects.
> Prior works achieve exponential rates with large/adaptive step-size after the training dynamics goes through an unstable phase of loss oscillations and their convergence guarantees are not anytime. In particular, Zhang et al. 2025(a, b) proves exponential rates after a burn-in phase of $1/\gamma^2$ by employing adaptive step sizes. Before reaching this burn-in time, they only provide upper bound on the loss at the average iterate and for $t< 1/\gamma^2$, this upper bound is of the form $L(\bar{\mathbf{w}}_t)\le\exp(c\eta/t)$ where $\bar{\mathbf{w}}_t$ is the average iterate at timestep $t$ and $c>0$. Moreover, Zhang et al. 2025(a, b) do not provide any discussion on obtaining exponential rates for SGD. To the best of our knowledge, Wu et al. (2024) is the only work that discusses SGD in the context of large step-size, however, they only provide the unstable phase analysis where they obtain $O(\eta/t)$ upper bound on the average loss. We propose a GD scheme with an *increasing non-adaptive step-size* scheme with which the training dynamics never enters unstable phase and the obtained stretched exponential rate of $O(\exp(-\Omega(t^{1/3})))$ is *anytime*. We have accordingly added additional details in the Related Works section.
>
> For SGD, we design local curvature based step-size that provides the first stretched exponential rate under lightweight and implementable step-size rules.

---

### Decision · Action_Editor_S67U · 2026-05-25

**Recommendation:** Accept as is

**Audience:**

Yes

**Audience Explanation:**

see above.

**Claims And Evidence:**

Yes

**Claims Explanation:**

The reviews are broadly supportive of acceptance. Reviewers agree that the authors have substantially addressed the main technical and presentation concerns raised in the initial reviews, including correcting the convergence terminology, resolving the proof issue around Theorem 3.5, clarifying the distinction between Adaptive SGD and Block Adaptive SGD, and adding useful supplemental empirical material. The paper is viewed as mathematically sound, clearly motivated, and relevant to the TMLR optimization community, and the remaining concerns about experimental breadth and baseline comparisons do not appear to undermine the paper’s value as a theory-focused contribution. I therefore recommend acceptance to TMLR. However, I do not recommend elevation to the NeurIPS/ICLR/ICML journal-to-conference track: the reviewers uniformly weakly oppose J2C elevation, primarily because the contribution, while solid and technically interesting, is viewed as insufficiently novel or differentiated from prior work to meet the higher bar for a top-tier conference-track presentation.